# Targeting RAS-driven human cancer cells with antibodies to upregulated and essential cell-surface proteins

Alexander J Martinko[1,2], Charles Truillet[3], Olivier Julien[1], Juan E Diaz[1], Max A Horlbeck[4,5], Gordon Whiteley[6], Josip Blonder[6], Jonathan S Weissman[4,5], Sourav Bandyopadhyay[7], Michael J Evans[3], James A Wells[1,4]*

[1]Department of Pharmaceutical Chemistry, University of California, San Francisco, San Francisco, United States; [2]Chemistry and Chemical Biology Graduate Program, University of California, San Francisco, San Francisco, United States; [3]Department of Radiology and Biomedical Imaging, University of California, San Francisco, San Francisco, United States; [4]Department of Cellular and Molecular Pharmacology, University of California, San Francisco, San Francisco, United States; [5]Howard Hughes Medical Institute, University of California, San Francisco, San Francisco, United States; [6]Cancer Research Technology Program, Frederick National Laboratory for Cancer Research, Frederick, United States; [7]Diller Family Comprehensive Cancer Center, University of California, San Francisco, San Francisco, United States

**Abstract** While there have been tremendous efforts to target oncogenic RAS signaling from inside the cell, little effort has focused on the cell-surface. Here, we used quantitative surface proteomics to reveal a signature of proteins that are upregulated on cells transformed with KRAS$^{G12V}$, and driven by MAPK pathway signaling. We next generated a toolkit of recombinant antibodies to seven of these RAS-induced proteins. We found that five of these proteins are broadly distributed on cancer cell lines harboring RAS mutations. In parallel, a cell-surface CRISPRi screen identified integrin and Wnt signaling proteins as critical to RAS-transformed cells. We show that antibodies targeting CDCP1, a protein common to our proteomics and CRISPRi datasets, can be leveraged to deliver cytotoxic and immunotherapeutic payloads to RAS-transformed cancer cells and report for RAS signaling status in vivo. Taken together, this work presents a technological platform for attacking RAS from outside the cell.

*For correspondence: jim.wells@ucsf.edu

## Introduction

RAS is a family of three ubiquitously expressed small GTPases found in all animal cell types. RAS is localized to the intracellular leaflet of the cell membrane where it serves as a major communication hub that relays extracellular growth factor-dependent signaling to as many as a dozen different intracellular signaling pathways, including the classically studied MAPK and PI3K pathways (**Downward, 2003**). Collectively, these pathways induce dramatic changes to cells including transcriptional reprogramming, promotion of cell survival, suppression of apoptosis, metabolic rewiring, promotion of proliferation, and increased cell invasiveness (**Pylayeva-Gupta et al., 2011**). Many of these phenotypes are well-known hallmarks of cancer survival (**Hanahan and Weinberg, 2011**). Nearly one third of all human malignancies are driven by mutational activation of one of the three RAS isoforms: KRAS, NRAS and HRAS (**Roberts and Der, 2007**). Hence, oncogenic RAS has been an important focus of the cancer biology and drug discovery communities for several decades (**Ledford, 2015**;

*Stephen et al., 2014*). In efforts to identify tractable drug targets in RAS-driven cancers, tremendous research emphasis has been placed on understanding oncogenic RAS and its role in the dysregulation of intracellular signaling pathways (*Papke and Der, 2017*). Despite these intense efforts to target intracellular pathways, little is understood about how RAS signaling can regulate the cell surface proteome, the surfaceome.

The surfaceome represents the dominant means by which cells communicate, interact, obtain nutrients, and engage the immune system. Overexpression of oncogenic RAS in model cell lines contributes to loss of adhesion, increased invasive properties, and evasion of immune responses, phenotypes that depend on the function of membrane proteins (*Pylayeva-Gupta et al., 2011*). This suggests that RAS-driven transcriptional reprogramming coordinately regulates the expression of cell surface proteins to exert malignant phenotypes. We believe systematic efforts are needed to identify the influence of oncogenic RAS on surface protein expression. Such studies would aid in progressing our fundamental understanding of how RAS drives malignancy. Additionally, they may lead to the identification of novel targets for next generation antibody and cellular therapy based intervention in RAS-driven cancers.

Here, we applied quantitative mass spectrometry (MS)-based proteomics on an isogenic diploid epithelial cell model, MCF10A (*Soule et al., 1990*), to measure the influence of oncogenic KRAS signaling on the expression of proteins in the surfaceome. We find that KRAS dramatically alters the expression of membrane proteins and coordinately regulates groups of proteins critical for managing cell adhesion and migration. Using pharmacological inhibitors to interrogate the contribution of individual effector pathways, we find that the oncogenic KRAS-induced changes in the surfaceome are driven predominantly through the MAPK effector pathway. From these discovery efforts, we derived a panel of recombinant antibodies to half a dozen of these differentially expressed targets, and applied the antibodies to confirm their surface expression patterns in a panel of mutant KRAS-driven cancer cell lines.

To further focus our efforts on those upregulated surface targets that are most critical for KRAS-driven cell growth, we conducted a CRISPRi screen to assess the relative functional importance of more than 1600 proteins in the oncogenic KRAS surfaceome as compared to control cells. We discovered KRAS-transformed cells have a selective dependence on the surfaceome to maintain both Integrin and Wnt signaling, pathways critical for regulation of adhesion and cell migration, respectively. One of these targets, C̲UB D̲omain C̲ontaining P̲rotein 1 (CDCP1), a protein known to drive loss of adhesion through integrin signaling (*Bhatt et al., 2005*; *Casar et al., 2014*), was a common hit in both our MS and CRISPRi screens. We focused on this example and showed that recombinant antibodies to CDCP1 can be used to selectively deliver therapeutic cytotoxic payloads or recruit and activate T-cells to pancreatic cancer cells while sparing normal control cells. We further show in mouse xenograft models for pancreatic cancer that CDCP1 expression level is an abundant biomarker for RAS signaling through the MAPK signaling pathway. Our studies provide a mechanism for how oncogenic KRAS can influence the expression and function of membrane proteins, and how this knowledge can be leveraged to identify potential targets to attack RAS from outside the cell.

## Results

### Characterization of the oncogenic KRAS surfaceome

The three major cancer types in which RAS mutations are the most prevalent (pancreatic, lung, and colorectal) all arise from epithelial tissue (*Stephen et al., 2014*). To begin to isolate the effects of KRAS transformation on the cell surfaceome, we chose a well-characterized, non-tumorigenic immortalized epithelial cell line, MCF10A, to generate an isogenic model for KRAS transformation (*Debnath et al., 2003*; *Martins et al., 2015*). The diploid MCF10A cell line is often used for oncogenic transformation studies because it is non-malignant and harbors only small genetic modifications typical of a culture adapted cell line. In addition, MCF10As are readily transduced with lentivirus, which enabled the rapid generation of the isogenic model as well as the possibility to engineer these cells further for downstream validation studies. Importantly, MCF10A cells stably transduced with oncogenic KRAS$^{G12V}$ undergo numerous phenotypic changes characteristic of malignant transformation, including increased proliferation and significant loss of cell adhesion (*Stolze et al., 2015*). While MCF10A cells are mammary in origin, and RAS is rarely implicated in

breast cancer, we believe that the benefits of this model outweighs this caveat. Moreover, we hypothesized RAS signaling in any epithelial cell type would result in a common transcriptional program and signature of cell-surface protein expression. Recently, we showed using initial proteomics discovery experiments that the surfaceome does indeed change substantially in MCF10A cells expressing oncogenic KRAS (*Ye et al., 2016*). However, to determine the therapeutic potential of exploiting differences in the surfaceome in RAS-driven cancers, we needed to quantitatively measure the extent to which these proteins change.

To measure changes in the cell surface proteome in an unbiased manner, we applied the recently established glycoprotein capture and enrichment proteomic method. This is advantageous because greater than 90% of surface proteins contain Asn-linked glycosylation (*Schiess et al., 2009*; *Wollscheid et al., 2009*). We adapted the method to quantitatively compare the surface protein levels on the two isogenic cell lines, MCF10A cells with or without oncogenic KRAS$^{G12V}$, using stable isotope labeling with amino acids in cell culture (SILAC) (*Figure 1A*) (*Ong and Mann, 2006*). Isotopically labeled cells were harvested and treated with sodium periodate to oxidize the vicinal diols on the cell surface glycans. Subsequently, cells were treated with biocytin-hydrizide to label the oxidized glycans with biotin for pull-down and enrichment. The proteins were captured on streptavidin beads, trypsinized, washed, and N-glycosylated peptides were released from the beads by N-glycanase (PNGase) treatment. Released peptides were then analyzed by LC-MS/MS.

We identified a total of 2943 tryptic peptides from ~500 proteins common in both the MCF10A cells with and without KRAS$^{G12V}$ (*Figure 1—figure supplement 1A*). Of 2943 tryptic peptides, 1460 contained the deamidated asparagine mass (0.984 Da) shift generated by N-glycanase treatment, providing high confidence it contained an asparagine-linked modification. Next, we eliminated proteins for which only a single peptide was identified, to distill to a higher confidence set. Lastly, the data were filtered bioinformatically to isolate proteins annotated in UniProt to be localized to the cell surface, resulting in 1257 high confidence peptides originating from 258 membrane-associated proteins found in both light and heavy samples. Biological replicates showed 88% overlap in the common membrane proteins identified, and good correlation in terms of SILAC ratios ($R^2$ = 0.7676) (*Figure 1—figure supplement 1B–C*). In total, we identified 17 proteins that were significantly upregulated, and 22 that were significantly downregulated (fold-change >±1.75; p-value<0.01) (*Figure 1B*; The full dataset can be found in *Supplementary file 1*). Gene ontology (GO) analysis of these significantly altered proteins in our datasets revealed significant enrichment for proteins involved in cell adhesion, cell motion, and cell-cell adhesion (*Figure 1—figure supplement 1D*).

We next wanted to understand how different signaling pathways emanating from RAS were contributing to the changes we observed in the surfaceome. We employed the same proteomics approach but instead examined the consequences of treatment with well-established pharmacological inhibitors, MEKi (PD0325901) and AKTi (MK2206) that block the two classic pathways downstream of RAS, MAPK and PI3K respectfully (*Barrett et al., 2008*; *Hirai et al., 2010*). As a control, we also tested the effect of Gefitinib, which inhibits EGFR, a receptor tyrosine kinase found upstream of RAS (*Moasser et al., 2001*). In each experiment, isotopically labeled MCF10A KRAS$^{G12V}$ cells were treated with subtoxic concentrations of drug for 72 hr and compared to non-isotopically labeled vehicle treated MCF10A KRAS$^{G12V}$ cells (*Figure 1A*). For the MEKi experiment, we were able to quantify the relative expression levels of 250 proteins (*Figure 1C*; The full dataset can be found in *Supplementary file 2*). Biological replicates showed similar overlap and reasonable correlation in fold-changes in expression ± MEKi ($R^2$ = 0.61), indicating biological and technical reproducibility (*Figure 1—figure supplement 1B–C*). Remarkably, 13 of the proteins that were significantly altered by KRAS$^{G12V}$ signaling were reversibly influenced by MEKi in the KRAS$^{G12V}$ cells (*Figure 1D*). By contrast, when the KRAS$^{G12V}$ cells were treated with AKTi or EGFRi, very few significant changes were observed (*Figure 1E* and *Figure 1—figure supplement 1E–F*; Full datasets can be found in *Supplementary file 3* for AKTi and *Supplementary file 4* for EGFRi). Others have shown the importance of the MAPK pathway for RAS-mediated transformation and tumerogenesis (*Roberts and Der, 2007*). Consistent with previous reports, our findings indicate that the MAPK pathway is the dominant pathway by which RAS mediates influence on the surfaceome in the MCF10A model. Taken together with our unbiased GO analysis, these results suggest a model wherein RAS signaling through the MAPK pathway promotes the coordinate expression of proteins that may contribute to increased invasiveness, metastasis, and epithelial to mesenchymal transition.

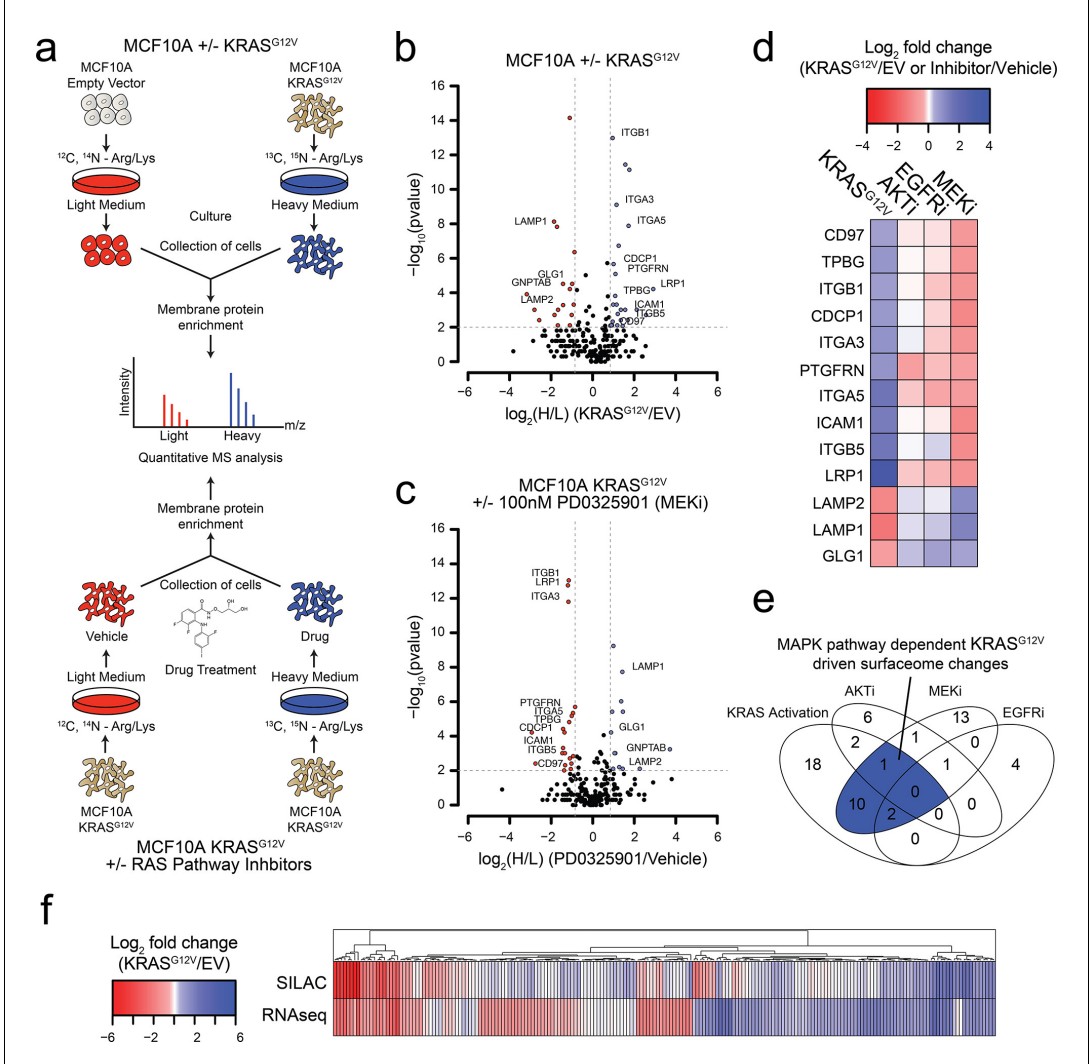

**Figure 1.** Oncogenic KRAS signaling coordinately regulates the expression of cell surface proteins in a model epithelial cell. (a) Experimental strategy for quantitative SILAC surface proteomics to compare surfaceomes of MCF10A empty vector cells to MCF10A KRAS[G12V] cells (Top), or MCF10A KRAS[G12V] cells treated with vehicle versus RAS pathway inhibitors (Bottom). Cells were cultured in either light or heavy SILAC media and then processed using N-linked cell surface glycoprotein enrichment and MS-based proteomic analysis. (b) Volcano plot of MCF10A empty vector versus MCF10A KRAS[G12V] cell-surface mass spectrometry experiment showing $\log_2$ fold-changes in expression (X-axis) or $-\log_{10}$(p-value)s (Y-axis). Proteins with a p-value less than 0.01 and a minimum of 1.75 fold increase (blue) or decrease (red) in SILAC ratio were considered significantly changed. Proteins labeled with text show expression patterns that were significantly altered by oncogenic KRAS signaling and inversely altered by MEKi. (c) Volcano plot representations of surface proteins in MCF10A KRAS[G12V] cells with or without treatment with the MEK inhibitor (MEKi), PD0325901 (100 nM). Proteins with a p-value less than 0.01 and a minimum of 1.75 fold increase (blue) or decrease (red) in SILAC ratio were considered significantly changed. Proteins labeled with text showed expression patterns that were significantly altered by oncogenic KRAS signaling and inversely altered by MEKi. (d) Heatmap representation of the SILAC mass spectrometry data for proteins that were significantly changed by both KRAS[G12V] and MEKi. (e) Venn diagram showing overlap of targets found in the SILAC mass spectrometry data for significantly altered proteins detected in all experiments. (f) Heatmap representation of a comparison between RNAseq and SILAC MS-based proteomics for all commonly identified genes. These data showing modest correlation highlight the importance of quantifying cell-surface proteomes by mass spectrometry for higher confidence identification of candidate proteins for antibody generation.

The online version of this article includes the following figure supplement(s) for figure 1:

**Figure supplement 1.** Oncogenic KRAS signaling coordinately regulates the expression of cell surface proteins in a model epithelial cell.

To further characterize the influence of oncogenic KRAS on the surface proteome, we performed RNAseq on both the MCF10A empty vector control and KRAS$^{G12V}$ cells. The correlation between expression level changes observed in the SILAC proteomic data with those from RNAseq data (*Figure 1F*; The full RNAseq dataset can be found in *Supplementary file 5*.) was modest but significant (R2 = 0.422). This modest correlation is not surprising as there are many possible points of regulation between synthesis of an mRNA and trafficking of a protein to the cell surface that can blur the connection between steady state RNA and protein levels. Others have reported a similar degree of correlation between levels of cytosolic proteins and their mRNAs (*Lundberg et al., 2010*; *Schwanhäusser et al., 2011*; *Wiita et al., 2013*). Nonetheless, the combination of the proteomic and transcriptomic data reinforces that KRAS transformation drives significant and coordinated changes in the cell surface. Additionally, we identified a large cluster of genes found to be upregulated in both data sets that provides a set of high-confidence KRAS signaling surface markers for further validation.

## Generation of recombinant antibodies that target the KRAS surfaceome

In order to orthogonally validate these results in a variety of cellular settings, we sought robust antibody tools to detect and study KRAS regulated surfaceome proteins. Unfortunately, recombinant monoclonal antibodies are not commercially available for the vast majority of cell surface proteins (*Wollscheid et al., 2009*). Moreover, others have shown that up to half of all commercially available antibodies are unreliable (*Baker, 2015*). Hence, we sought to generate and validate recombinant monoclonal antibodies using antibody phage display for a set of the most interesting KRAS-induced targets. Recombinant antibodies are particularly useful as they are renewable, thus creating a reliable resource. Furthermore, they can be bioengineered for many useful applications including therapeutic payload delivery and in vivo imaging. We leveraged the quantitative proteomic and RNAseq data to manually choose seven high-confidence KRAS-induced single-pass transmembrane receptors to advance to a recombinant antibody engineering campaign. We chose single-pass receptors as they are more amenable to antibody discovery using soluble extracellular domains (ECD) and were well-represented in our data. All candidates were at least 2-fold upregulated as observed by mass spectrometry in the KRAS$^{G12V}$ cells as compared to empty vector. In most cases, the candidates were also at least 2-fold upregulated in the same context as determined by RNAseq. The representative targets had a range of functions critical for tumor survival. For example, CDCP1, ICAM1, and ITGB5 all play roles in cell adhesion and migration (*Huttenlocher and Horwitz, 2011*; *Kevil et al., 2004*); NT5E is a nucleotidase that contributes to immune cell evasion (*Antonioli et al., 2016*); LTBR is a cytokine receptor involved in maintenance of lymphoid microenvrionments (*Lukashev et al., 2006*); CD55 is a complement system inhibitor (*Sun et al., 1999*), and ANPEP is a protease thought important for metastasis (*Sørensen et al., 2013*).

To enable the rapid expression and purification of these target proteins for generation of antibodies by phage display, we expressed their ECDs as Fc-fusion proteins in mammalian cells (*Figure 2A*) (*Czajkowsky et al., 2012*). We introduced a biotin-acceptor-tag at the C-terminus, and a TEV proteolysis site between the ECD and Fc-domain. These tags allowed for site-selective capture of the ECD-Fc fusion on magnetic streptavidin beads, and release of each ECD containing bound Fab-phage after TEV treatment. This 'catch-and-release' strategy ensured selective release of Fab-phage bound to each ECD, while avoiding enrichment of unwanted Fab-phage that either bound the Fc-domain or the streptavidin beads (*Hornsby et al., 2015*) (*Figure 2B*). To facilitate production of the biotinylated construct, a stable HEK293T cell line was engineered to overexpress the bacterial biotin ligase, BirA, with an ER retention tag. This allowed for site-specific biotinylation to be performed in cells, thus eliminating the need for in vitro labeling. Western blots confirmed that these cells expressed and secreted quantitatively (typically >90%) biotinylated Fc-fusion proteins (*Figure 2—figure supplement 1A*). Each Fc-fusion protein was transiently expressed in the engineered HEK293T cells and purified by Protein A affinity purification from the media for use in phage-antibody selections. We conducted 4 rounds of 'catch-and-release' selections with a well-validated synthetic Fab-phage library (*Persson et al., 2013*). After each selection, we isolated 95 individual phage clones and screened them for target binding by phage ELISA followed by DNA sequencing (*Figure 2C* and *Figure 2—figure supplement 1B*). Selections resulted in the identification of 116 unique antigen-binding sequences against the seven ECD-Fc fusion targets (*Figure 2—figure*

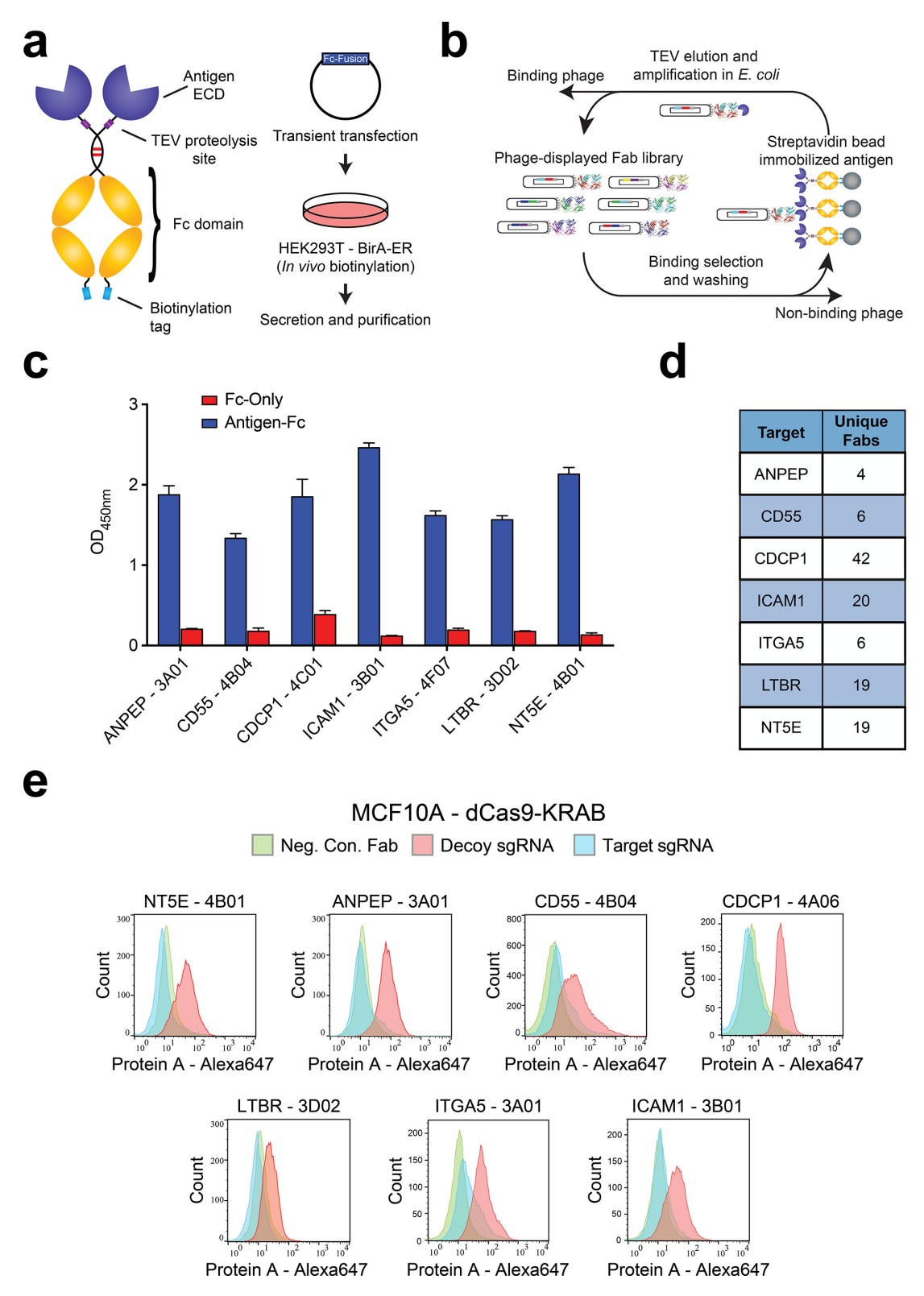

**Figure 2.** Generation and validation of antibodies to oncogenic KRAS upregulated surface proteins. (a) (Left) Schematic of the Fc-fusion construct developed for rapid expression of membrane protein extracellular domains. Each extracellular domain was expressed as a TEV cleavable site-specifically biotinylated Fc-fusion. (Right) HEK293T cells stably expressing an ER-localized biotin ligase are transiently transfected with the Fc-fusion expression vector. Proteins are quantitatively biotinylated in-vivo, secreted into the cellular media, and purified by Protein A affinity purification. (b)

*Figure 2 continued on next page*

*Figure 2 continued*

Shown is the strategy for phage display generation of antibodies to each RAS-induced protein ECD. Proteins were immobilized on streptavidin magnetic beads and mixed with a highly diverse phage-displayed Fab library. Non-binding phage were removed by washing and phage bound protein was released by enzymatic treatment with TEV protease. Eluted phage were propagated in *E. coli* and the selection process was iterated for 3–4 rounds to enrich the library for specific protein binders. (c) Representative phage ELISAs from selections against seven proteins seen elevated in expression level by oncogenic KRAS signaling in MCF10As. Phage clones show strong binding to cognate protein Fc-fusions but little detectable binding to the isolated Fc-domain suggesting binding to the targeted ECD. (d) Table of the number of unique antibody clones generated against each of the specified KRAS upregulated targets. (e) Representative flow cytometry histograms demonstrate specific cellular target engagement of Fab clones generated against seven KRAS-driven surface proteins. MCF10A cells stably expressing dCas9-KRAB and a decoy sgRNA (red) or target sgRNA (blue and green) were labeled with either a negative control Fab (green) or a Fab of interest (red and blue). Fab binding to cells was detected by addition of a Protein A Alexa647 conjugate and quantification by immunofluorescence flow cytometry.

The online version of this article includes the following figure supplement(s) for figure 2:

**Figure supplement 1.** Generation and validation of antibodies to oncogenic KRAS upregulated surface proteins.

*supplement 1D*). None of these clones demonstrated appreciable binding to the Fc-domains fused to the antigen, a testimony to the efficiency of the catch-and-release strategy. For further characterization, each Fab was cloned into an *E. coli* secretion plasmid and expressed, typically in yields ranging from 1 to 10 mg/L. Fabs were purified from the periplasm by Protein A purification for further analysis.

To validate the antibodies, we adopted several of the tests recently recommended by the Working Group for Antibody Validation (*Uhlen et al., 2016*). Firstly, we generated a stable cell line for each target that overexpressed the protein ECD fused to a fluorescent protein expression reporter and a generic single-pass transmembrane domain (*Figure 2—figure supplement 1C*). Selections to each of the seven targets produced multiple antibodies showing dramatically increased binding to cells over-expressing the target ECD as compared to control cells (*Figure 2—figure supplement 1D*). We further validated the specificity of the antibodies using CRISPRi knockdown of each target in the MCF10A KRAS$^{G12V}$ cells (*Gilbert et al., 2013*). In each case, we identified at least one antibody that showed significant staining on the MCF10A KRAS$^{G12V}$ cells and little to no binding for the CRISPRi knockdown corresponding cell line (*Figure 2E*). Importantly, these data also corroborated the proteomics observation that these membrane proteins are highly expressed in the MCF10A KRAS$^{G12V}$ cells.

## Validation of oncogenic KRAS-induced membrane proteins

To orthogonally validate the observations made by MS-based proteomics and RNAseq analysis, we measured the relative cell-surface abundance of each protein by flow cytometry using our recombinant antibodies. Gratifyingly, all seven proteins showed elevated expression on the MCF10A KRAS$^{G12V}$ cells relative to the empty vector control (*Figure 3A*). Moreover, the flow cytometry data generated with the Fabs generally correlated well with the proteomic measurements. These results were further confirmed by immunofluorescence (IF) for a subset of the targets, where we observed both differential expression and cell surface localization in each case (*Figure 3B*).

Next, we expanded beyond the isogenic MCF10A cell model to measure the relative expression of the target proteins on a panel of tumorigenic cells that included lung, colorectal, and pancreatic cancer cell lines known to harbor oncogenic RAS mutations (*Figure 3C*). These three cancer types have the highest occurrence of oncogenic KRAS. For five of the seven targets (CDCP1, CD55 NT5E, ICAM1 and LTBR) we observed high levels of expression in at least half of the eight cancer cell lines. Most strikingly we observed that one protein, CDCP1, was ubiquitously abundant in expression across the entire panel. In contrast, two of the proteins, ANPEP and ITGA5, showed little detectable expression in all cases. In general, variability of expression should not be surprising, as these cells vary in tissue origin and also harbor other cancer driving mutations beyond oncogenic RAS. While ANPEP and ITGA5 may be present if we were to screen a larger panel of cells, another possibility is that they are proteins specific to the MCF10A cell surface. A HEK293T control showed no expression of any of the targets.

We next tested by immunostaining how RAS pathway inhibitors would affect the expression of these seven target proteins. Indeed, most of these proteins (CD55, NT5E, CDCP1, and ICAM1) had expression levels that were affected by KRAS pathway perturbation, while inhibition upstream of

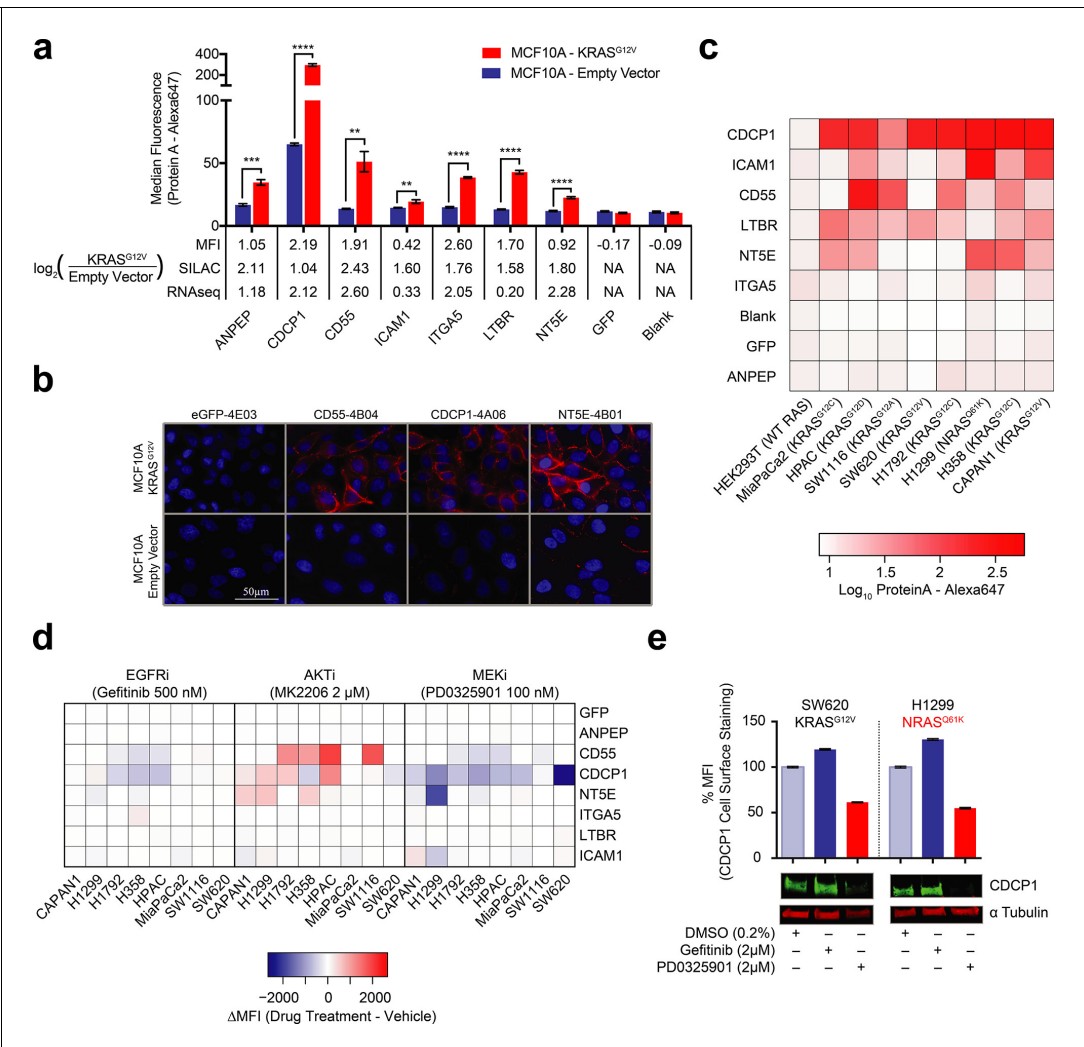

**Figure 3.** Validation of oncogenic KRAS-induced cell surface proteins with recombinant antibodies. (a) Seven proteins were confirmed to be upregulated on the MCF10A KRAS[G12V] cells by flow cytometry using specific recombinant antibodies ($n = 3$, error bars represent s.d.). The table below compares log$_2$ fold-changes as measured by flow cytometry, SILAC proteomics, and RNAseq. (b) Representative immunofluorescence images demonstrate orthogonal confirmation of KRAS-driven differential expression and cell surface localization of target proteins. (c) Heatmap representation of flow cytometry median fluorescent intensity values for Fabs to seven upregulated targets plus a Fab to GFP as a control for binding against nine tumorigenic cell lines. (d) Heatmap representation of flow cytometry median fluorescent intensity values for the eight Fabs against nine tumorigenic cell lines treated with RAS pathway inhibitors. (e) Flow cytometry and western blot profiling of CDCP1 expression levels in response to RAS pathway inhibitors.

KRAS by EGFRi had little effect in most cases (*Figure 3D*). Interestingly, AKTi inhibition caused increased expression of CDCP1, NT5E, and CD55 in multiple cases. A potential explanation is that MK-2206 inhibition resulted in feedback-driven MAPK signaling as has been previously observed (*Iida et al., 2013*). Also notable was that MEK inhibition resulted in highly decreased CDCP1 expression levels in nearly every cell line tested. This is consistent with our observations in the MCF10A cells and with previous reports in mutant RAS non-small cell lung cancer cells(*Uekita et al., 2014*). We further validated this result by western blot, demonstrating that MAPK signaling drives both CDCP1 expression and cell-surface localization (*Figure 3E*). Taken together, our antibody profiling provides corroborative evidence that oncogenic RAS influences the expression of specific membrane proteins at the cell surface. We believe this collection of highly specific recombinant antibodies constitute an expandable and enriched tool kit that may be useful for interrogating RAS-transformed cells and tissues.

## Functional characterization of the KRAS surfaceome in MCF10As using a CRISPRi screen

In addition to identifying proteins upregulated by expression of oncogenic KRAS, we explored how oncogenic KRAS signaling can influence how cells depend on specific surface proteins. We tested the impact on cell proliferation of selective knockdown of more than 1600 annotated membrane proteins using CRISPRi (*Figure 4A*; The sequences of all sgRNAs in the library can be found in *Supplementary file 6*). These target proteins included nearly all of those found in our mass spectrometry screen. We introduced the stable expression of dCas9-KRAB into both the wild type and KRAS$^{G12V}$ MCF10A cells using well-established constructs and standard lentiviral transduction (*Gilbert et al., 2014*). We transduced a surfaceome targeted lentiviral sgRNA library consisting of 5 sgRNAs per gene and 150 non-targeting controls into either the KRAS$^{G12V}$ MCF10A or the parental cell line. Cells were grown for 12 days, and at least 10 doublings. The MCF10A cells require the addition of growth factors (EGF, Insulin) to proliferate, whereas KRAS$^{G12V}$ MCF10A cells are growth factor independent. Thus, we screened the KRAS$^{G12V}$ MCF10A cells with and without growth factors in an attempt to maintain KRAS addiction while knocking down genes. After conducting the three screens, cells were harvested and genomic DNA was isolated. The sgRNA barcodes were PCR amplified and subjected to next generation sequencing (NGS) to quantify the effect of each sgRNA on growth rate. Genes were scored based on the average phenotype of the three strongest sgRNAs and p-values were calculated by a Mann-Whitney test against a non-targeting control set as previously described (*Gilbert et al., 2014*; *Kampmann et al., 2013*). Remarkably, individual knockdown of more than fifty genes caused a significant reduction (phenotype<-1 and p-value<0.05) in growth rate in the KRAS$^{G12V}$ cells grown without growth factor, while only four genes caused a significant growth advantage when knocked down (*Figure 4B*; The full datasets for all three screening conditions can be found in *Supplementary file 7*). Interestingly, there were only 12 genes in the wild type MCF10A cells and eight genes in the KRAS$^{G12V}$ cells grown with growth factor that showed a growth disadvantage, and none that were increased. We tested individual sgRNAs derived from the screening library for several genes in both cell lines and the results robustly confirmed the screening results (*Figure 4—figure supplement 1*).

In addition to the membrane protein genes targeted in our sgRNA library, we also included a set of known RAS pathway genes (*Figure 4C*). Not surprisingly, EGFR was critical for growth of wild type MCF10A grown in the presence of EGF. However, it was not critical to the KRAS$^{G12V}$ cells grown either in the presence or absence of growth factors, which reflects the fact that constitutively active KRAS$^{G12V}$ can drive the growth of these cells independent of EGFR signaling. Furthermore, the KRAS$^{G12V}$ cells showed a strong dependence on proteins within the MAPK signaling pathway (RAF/MEK/ERK) and little to no dependence on the PI3K/AKT pathway, consistent with previous reports highlighting the importance of RAS-driven MAPK signaling (*Roberts and Der, 2007*). Interestingly, RALGDS, a lesser-studied RAS effector protein, was of intermediate importance relative to MAPK and PI3K. Taken together, these results provide mechanistic insight as to how oncogenic KRAS drives signaling in MCF10A transformed cells. These self-consistent data in MCF10A cells recapitulate many known features of oncogenic RAS signaling, and rationalize the CRISPRi screening results. The set of genes that are critical for the growth factor independent proliferation of KRAS$^{G12V}$ cells but unimportant for the growth of wild type cells notably fell into two main categories (*Figure 4C*). One group constitutes core components of integrin-mediated cell adhesion, while the other included major proteins involved in Wnt signaling. Recently, others have observed an essential role for Wnt signaling in KRAS malignancies, a finding which our data recapitulate (*Wang et al., 2015*). Additionally, the CRISPRi results suggest a previously underappreciated dependency on integrin signaling proteins at the cell surface for growth factor independent proliferation driven by oncogenic KRAS.

Remarkably, there was very little overlap for proteins upregulated by KRAS as measured by mass spectrometry, and those showing synthetic growth reduction with KRAS in the CRISPRi screen (*Figure 4D*). This is not surprising since differential expression and acquired essentiality are different properties. However, two genes were identified as hits in both experiments, ITGB5 and CDCP1. CDCP1 has been previously implicated as an elevated marker in solid cancers and a driver of cancer cell growth, metastasis, and tumor progression (*Ikeda et al., 2009*; *Leroy et al., 2015*). CDCP1 was also observed in our earlier discovery proteomics screen using several membrane enrichment

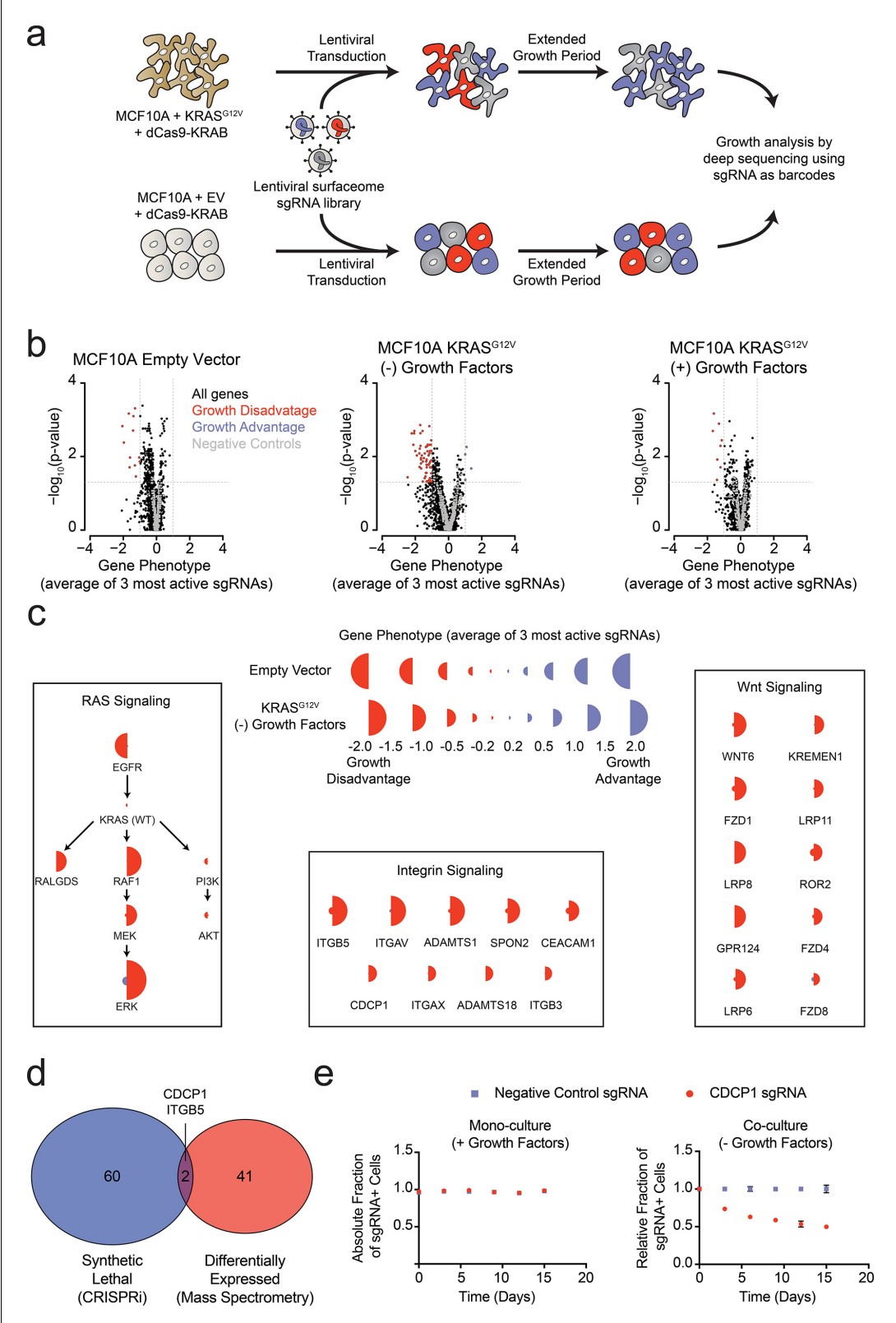

**Figure 4.** Systematic characterization of the functional importance of the KRAS surfaceome. (a) Experimental strategy for the CRISPRi surfaceome screen. MCF10A Empty Vector or MCF10A KRAS[G12V] cells were transduced with a lentiviral surfaceome sgRNA library containing five guides each for 1605 transmembrane targets. After being cultured for an extended growth period, the abundance of each sgRNA barcode was quantified by deep sequencing. (b) Volcano plots of the CRISPRi experiment showing effects on growth rate (gene phenotype,X-axis) versus –log$_{10}$(p-value)s (Y-axis).

*Figure 4 continued on next page*

*Figure 4 continued*

Proteins with a p-value less than 0.05 and a gene phenotype >1 (blue) or <-1 (red) were considered significant. (c) Shown is a representation of CRISPRi phenotypes for RAS pathway control genes (Left), functionally categorized KRAS[G12V] synthetic lethal genes including integrin signaling genes (Middle), and Wnt Signaling genes (Right). KRAS[G12V] synthetic lethal genes were defined as having a p-value<0.025 and a phenotype >0.8 for KRAS[G12V] cells and <0.8 for Empty Vector cells. (d) A Venn diagram comparing KRAS[G12V] synthetic lethal genes and differentially expressed genes. (e) Competitive growth assays performed with KRAS[G12V] cells transduced with the indicated CDCP1 and negative control sgRNAs. Cells were grown in monoculture with growth factors to ensure they maintained sgRNA expression (Left), and in co-culture with cells lacking sgRNA expression without growth factor to determine relative growth rates (Right). (*n* = 3, error bars represent s.d.).

The online version of this article includes the following figure supplement(s) for figure 4:

**Figure supplement 1.** Competitive growth assays performed with KRAS[G12V] (Left) and Empty Vector (Right) MCF10A cells transduced with the indicated sgRNAs.

methods (*Ye et al., 2016*). Interestingly, several groups have demonstrated that CDCP1 signaling can be mediated by interactions with integrins (*Casar et al., 2014*; *Wright et al., 2016*). Using an individual sgRNA to CDCP1, we observed a ~40% reduction in growth over 15 days of KRAS[G12V] cells with CDCP1 knocked down (*Figure 4E*). We confirmed that CDCP1 was indeed >98% reduced in expression by antibody staining.

While our results provide a large number of interesting proteins for follow-up, we decided to focus on targeting CDCP1 with our validated recombinant antibodies. Our antibodies did not appear to functionally inhibit CDCP1 (data not shown), however, we were motivated by the fact that we find it over-expressed in many RAS-driven cell lines. Moreover, its importance to growth makes it unlikely to be down-regulated as a resistance mechanism. These qualities suggest that CDCP1 could be a particularly attractive target for an antibody drug conjugate (ADC).

## Antibodies can selectively deliver toxic and immunotherapy payloads to mutant KRAS pancreatic cancer cells

Oncogenic KRAS is nearly ubiquitously expressed in pancreatic cancers and these cancers are particularly dependent on MAPK pathway signaling (*Eser et al., 2014*). Thus, we profiled CDCP1 expression levels on a panel of human Pancreatic Ductal Adenocarcinoma Cancer (PDAC) cell lines, and observed a high level of expression of CDCP1 relative to no observable expression on non-tumorigenic pancreatic duct cells, HPNE (*Figure 5A*). Indeed, quantification of the protein copy number of CDCP1 on the HPAC PDAC cells revealed that these cells were expressing ~2.9 million copies per cell (*Figure 5—figure supplement 1A*). Such high level of expression is rare, and is an important consideration for therapeutic targeting by ADCs (*Bornstein, 2015*). Moreover, cells containing oncogenic RAS are known to be very active in protein uptake by macropinocytosis and lysosomal degradation (*Commisso et al., 2013*), making an ADC strategy even more attractive.

We tested the ability of one of our recombinant CDCP1 antibodies to selectively deliver a cytotoxic payload to mutant KRAS tumor cells. Having the recombinant form facilitated simple conversion of the Fab to an IgG1 for the experiment. The HPAC or non-tumorigenic HPNE cells were treated with CDCP1 IgG1 that can bind a commercially available secondary antibody coupled to a non-cell permeable cytotoxic microtubule inhibitor, Monomethyl Auorstatin F (MMAF) (*Figure 5B*). We found that while the normal HPNE cells were unaffected by treatment of up to 20 nM of antibody complex, oncogenic KRAS containing HPAC cells were sensitive to ADC treatment at concentrations as low as 10pM. (*Figure 5C*). More than 60% reduction of viability was observed at 0.8 nM (*Figure 5D*), suggesting that the CDCP1-selective antibody can selectively internalize and deliver the cytotoxic payload. Similar levels of toxicity were observed in two additional mutant RAS cancer cell lines (*Figure 5—figure supplement 2*). It is not surprising that all cells were not killed since these are rapidly dividing cells in culture that create a steady state of apoptosis and proliferation. It is also possible that directly coupling MMAF to the recombinant antibody could further improve its potency as an ADC.

Recently, the FDA approved the first example of a bispecific T-cell engager (BiTE) antibody immunotherapy that binds both a tumor selective antigen, and CD3 on the surface of cytotoxic T-cells (*Wu et al., 2015*). The BiTE recruits the T-cell to the tumor cell, induces T-cell activation, and tumor cell killing. Many similar BiTEs are currently being tested in the clinic (*Yuraszeck et al., 2017*). We hypothesized that the CDCP1 recombinant Fab could be engineered in a BiTE modality to

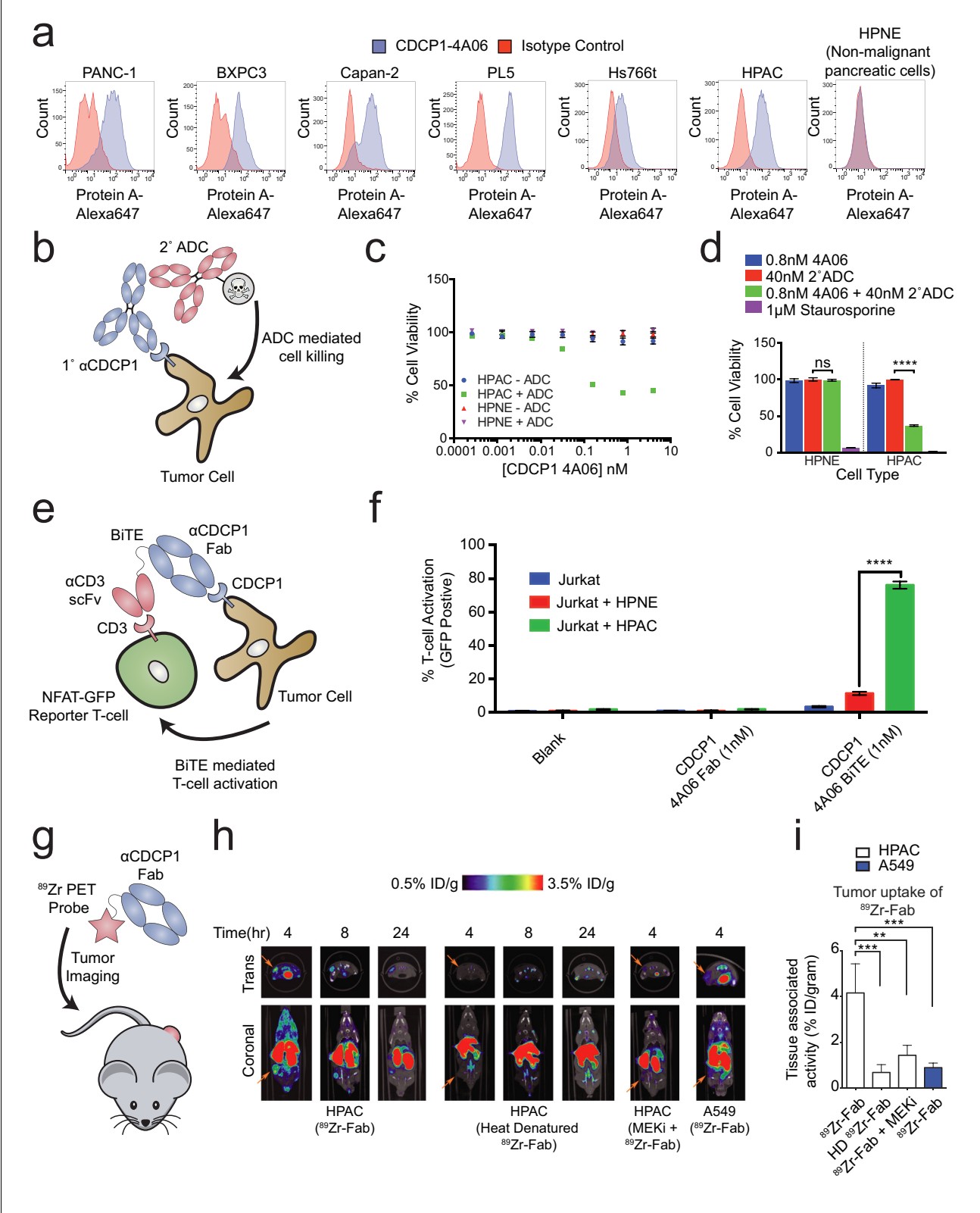

**Figure 5.** Systematic approaches to demonstrate the utility of the recombinant antibodies to target and detect RAS signaling in pancreatic ductal adenocarcinoma (PDAC). (a) Profiling of a panel of pancreatic ductal adenocarcinoma cells by flow cytometry demonstrates that CDCP1 is highly expressed on PDAC cells. Remarkably, CDCP1 was expressed at much lower levels on non-tumorogenic cells derived from the same tissue origin. (b) A schematic representation of the antibody drug conjugate cell-killing assay. Cells were treated with a primary IgG that targets CDCP1 and a secondary

*Figure 5 continued on next page*

Figure 5 continued

anti-human IgG conjugated to the cytotoxic drug monomethyl auristatin F (MMAF). Cellular viability was quantified by CellTiter-glo after 72 hr incubation with antibody treatment. (c) Dose-dependent antibody drug conjugate-mediated cell killing was only observed in the HPAC tumorigenic cells and not in the non-tumorigenic HPNE cells (n = 3, error bars represent s.d.). (d) Sub-nanomolar treatment with a CDCP1 IgG was sufficient to selectively kill greater than 50% of HPAC cells, but only when in combination with a stoichiometric excess of the secondary antibody drug conjugate (n = 3, error bars represent s.d.). (e) Schematic of the experimental setup for the flow cytometry-based T-cell activation assay used to profile BiTE activity. Cells were incubated with HPAC or HPNE target cells in the presence or absence of antibody treatment. After overnight incubation, T-cell activation was quantified via the expression of an NFAT-dependent GFP reporter gene. (f) Jurkat cells were significantly activated when treated with 1 nM BiTE in the presence of HPAC target cells as compared to HPNE control cells. Importantly, treatment with the CDCP1 BiTE alone or with Fab lacking the CD3 targeting component resulted in no significant T-cell activation. (g) A schematic representation of the in vivo PET imaging study. $^{89}$Zr-labeled CDCP1 Fabs were used for PET imaging of PDAC xenograft bearing mice. (h) Representative microPET images of four immunocompromised nu/nu mice bearing cancer xenografts targeted with a $^{89}$Zr-labeled CDCP1 Fab. Images over time show the tumor specific expression of CDCP1, as well as the persistent binding of the Fab to the tumor over 24 hr (Left). Importantly, when the same Fab was heat denatured prior to injection or when a negative control xenograft was used, there was no observable uptake of the $^{89}$Zr-Fab (Middle and Right). Remarkably, no uptake was observed in the mouse treated with a sub-toxic dose of MEKi prior to imaging, demonstrating the coupling of CDCP1 expression MAPK pathway signaling in vivo. (i) Quantification of tumor specific bio-distribution of the CDCP1 $^{89}$Zr-Fab in tumor-bearing nu/nu mice (n = 5 per treatment arm) confirms the trends observed by microPET imaging. Tumor localization of $^{89}$Zr-Fab was antigen dependent and ablated by specific inhibition of MAPK signaling.

The online version of this article includes the following figure supplement(s) for figure 5:

**Figure supplement 1.** Characterization of CDCP1 expression levels in cancer cell lines and normal human tissues.
**Figure supplement 2.** Application of recombinant antibody drug conjugates to target RAS signaling in cancer.
**Figure supplement 3.** CDCP1 expression is coupled to MAPK signaling in mice.

selectively target oncogenic KRAS PDAC cells. We genetically fused the CDCP1 Fab to the clinically utilized OKT3 anti-CD3 scFv, and tested whether it could mediate T-cell activation with an NFAT-GFP reporter Jurkat T-cell in co-culture (*Figure 5E*). Remarkably, 1 nM of the CDCP1 BiTE was sufficient to activate 75% of T-cells when co-cultured with mutant KRAS HPAC cells, while co-culture with HPNE cells, or treatment with BiTE lacking the anti-CD3 scFv, resulted in minimal activation (*Figure 5F*).

## Application of antibodies for in-vivo detection of oncogenic RAS-dependent MAPK signaling

As a final proof of concept application of our antibodies, we tested to see whether CDCP1 expression level could be used as an in vivo imaging marker for RAS signaling in a mouse xenograft model for pancreatic cancer (*Figure 5G*). The CDCP1-4A06 Fab was labeled with a positron-emitting radio-isotope $^{89}$Zr and used for positron-emission-tomography (PET) imaging in mice with mutant RAS HPAC xenographs. As a negative control, $^{89}$Zr-Fab was heat denatured prior to administration. As an additional control, we used xenograft mouse models with A549 cells that express only 60,000 copies of CDCP1 per cell (*Figure 5—figure supplement 1A*). Although A549 cells also harbor an oncogenic RAS mutation, previous work has shown that growth and MAPK signaling in these cells are not significantly diminished by knockdown of KRAS (*Singh et al., 2009*). Within four hours of administration of the $^{89}$Zr-Fab, tumor localization was observed, and by eight hours staining was highly pronounced (*Figure 5H*). Significantly reduced tumor staining was observed using the heat denatured $^{89}$Zr-Fab. Strikingly, in the HPAC xenograft mice treated with subtoxic MEKi, $^{89}$Zr-Fab tumor localization was also highly diminished. These animal data provide further evidence to the hypothesis that CDCP1 expression is a direct consequence of RAS dependent MAPK signaling even in the complex milieu of a tumor. Excised tumors were used to quantify PET signal, confirming the trends seen in the qualitative images (*Figure 5I* and *Figure 5—figure supplement 3*)

## Discussion

Exploiting an unbiased approach of globally and quantitatively profiling the oncogenic RAS surfaceome that relies on MS-based proteomics and CRISPRi targeted screening, we begin to demonstrate the logic of how constitutive RAS activation influences the expression and function of membrane proteins to exert oncogenic phenotypes. Our results suggest that RAS signaling can coordinately regulate a network of surface proteins in order to drive malignant hallmarks such as loss of adhesion, cell migration, and cell growth. Specifically, our work reveals that proteins involved

in integrin signaling are enriched both in terms of function and expression level at the RAS cell surface. For example, we observed that cells with oncogenic KRAS display high expression levels of several integrins (ITGB5/ITGB1/ITGA5/ITGA3) in combination with CDCP1. Previous work has identified a signaling axis through which CDCP1/integrin complexes can drive AKT dependent loss of adhesion, cell growth, and anti-anoikis signaling (*Casar et al., 2014*). We found that knockdown of CDCP1 or ITGB5 slows the growth of cells addicted to RAS signaling, supporting the idea that oncogenic RAS promotes the expression of these proteins to enable tumor cell growth. This also suggests that the invasive potential and propensity for metastasis of RAS-driven cancers may partially be a consequence of these differentially expressed proteins working in concert. Furthermore, the evidence for this signaling axis is found in the TCGA for pancreatic cancer, where CDCP1 and all four identified integrins show significantly elevated mRNA levels in cancers with RAS mutations.

Activation of RAS can lead to signaling through a dozen effector pathways; however, our work suggests that the MAPK pathway is most important for regulating the surface proteome. Nearly half of the proteins we observed to be enriched on the surface of KRAS-transformed MCF10As were decreased in expression level when cells were treated with a MEK inhibitor. Conversely, treatment with EGF and AKT inhibition had little consequence on this set of RAS enriched membrane proteins. Consistent with this observation, we found the same cells were sensitive to knockdown of proteins in the MAPK pathway, but insensitive to CRISPRi knockdown of proteins involved in PI3K signaling. We used CDCP1 as an example to validate that these observations extend beyond MCF10As to patient derived cancer cell lines and even to xenograft mouse models for PDAC. These data provide mechanistic insight from the perspective of cell surface proteins to support previous reports that the MAPK pathway is the most potent of the RAS effector pathways in mediating transcriptional reprogramming and oncogenesis. Furthermore, our data suggest that a subset of these proteins could serve as new biomarkers for both RAS dependent MAPK signaling and drug efficacy of MAPK pathway inhibitors.

Our follow up investigation of CDCP1 exemplifies the utility of our technological pipeline, validates our approach of studying RAS signaling in a simple isogenic MCF10A model, and most importantly, provides a validated example of a cell surface marker for oncogenic RAS signaling in multiple cancer types. Our initial observation that CDCP1 expression was driven by RAS dependent MAPK signaling in MCF10As was recapitulated in more than a dozen mutant RAS cancer cell lines as well as an in vivo model for pancreatic cancer. While previous work has suggested a connection between RAS signaling and CDCP1 in lung cancers (*Uekita et al., 2014*), our work now demonstrates MAPK signaling-dependent expression of this protein in multiple epithelial-derived RAS-driven malignancies as well as in vivo. Additionally, we show that CDCP1 is important for proliferation in RAS-addicted cells, suggesting a significant link between its function and its upregulated surface expression. Here, we applied our novel antibodies to CDCP1 to begin to validate its potential as a therapeutic target and as a biomarker in RAS-driven cancers dependent on MAPK signaling.

More preclinical studies beyond the scope of this work are necessary to fully validate CDCP1 as a therapeutic target. Nonetheless, we saw tumor localization of our CDCP1 antibodies in mouse xenograft models for PDAC, and MEK-dependent expression providing mechanistic validation in vivo. This mouse experiment does not provide a fair assessment of tumor selective expression of CDCP1, as our antibody did not show cross reactivity with the mouse homolog of CDCP1 (data not shown). However, analysis of the GTEX tissue expression database reveals that CDCP1 normal tissue expression is quite restricted (*Figure 5—figure supplement 1B*). In comparison to the classical antibody-targeted cell-surface protein, HER2, normal tissue expression for CDCP1 is far lower (*Figure 5—figure supplement 1C*). Additionally, we showed that mutant RAS PDAC cells express nearly 3 million copies per cell, a number comparable to the estimated 2 million copies per cell that can be observed in HER positive breast cancers. More extensive expression profiling of human tissues is needed to truly estimate the therapeutic potential of this protein. However, we believe that our collective observations provide motivation for the further exploration of CDCP1 an antibody target for RAS mutant cancers.

In summary, the results described herein validate and provide a mechanistic basis for the hypothesis that RAS signaling can meaningfully remodel the surface proteome to support malignant phenotypes. The strength of our linked target discovery and antibody generation pipeline was exemplified in the engineering of recombinant antibodies to CDCP1, and in demonstrating that these antibodies can be used to selectively deliver therapeutic payloads to RAS-transformed cells and report in vivo

on RAS signaling status. Overall, this work presents a novel technological pipeline for the systematic discovery and generation of antibodies to surface markers coordinately regulated by oncogene signaling pathways. Ultimately, we hope this pipeline can be applied towards the discovery of therapeutic antibodies to target previously 'undruggable' oncogenes, including RAS, from the outside.

## Materials and methods

### Cell surface protein enrichment

For all stable isotope labeling with amino acids in cell culture (SILAC) experiments, cells were cultured in DMEM SILAC media (Thermo) containing L-[$^{13}C_6$,$^{15}N_2$]lysine and L-[$^{13}C_6$,$^{15}N_4$] arginine (heavy label) (CIL) or L-[$^{12}C_6$,$^{14}N_2$]lysine and L-[$^{12}C_6$,$^{14}N_4$]arginine (light label) for seven doublings to ensure full incorporation of the isotopic labels. In the KRAS isogenic study, MCF10A cells expressing KRAS$^{G12V}$ were heavy labeled while the MCF10A cells harboring empty vector were light labeled. In all drug treatment experiments, light labeled cells were treated with DMSO while the heavy labeled cells were treated with 100 nM PD0325901, 2 µM MK-2206 (Selleck), or 500 nM Gefitinib (LC Laboratories). After 72 hr of drug treatment, cells were lifted with Versene (0.04% EDTA, PBS pH 7.4 Mg/Ca free) and 7–12 M cells from both the heavy and light-labeled cultures were mixed at a 1:1 cell count ratio for cell surface capture and enrichment (Wollscheid). Briefly, live cells were treated with sodium periodate buffer (2 mM NaPO$_4$, PBS pH 6.5) at 4°C for 20 min to oxidize vicinal diols on the glycans of surface glycoproteins. Aldehydes generated by oxidation were then covalently labeled with biocytin hydrazide in a labeling buffer (PBS pH 6.5 + 1 mM biocytin hydrazide (Biotium) +10 mM aniline (Sigma)) for 90 min at 4°C. Cells were then washed four times with PBS pH 6.5 to remove excess biocytin-hydrazide and stored overnight at −80°C.

Frozen cell pellets were thawed and lysed in RIPA buffer (VWR) containing 1X Protease Inhibitor Cocktail (Sigma) at 4°C for 30 min. Cell lysates were sonicated, clarified, and subsequently incubated with 500 µl of NeutrAvidin Agarose (Thermo) slurry at 4°C for 30 min. The NeutrAvidin beads were then extensively washed with RIPA buffer, high salt buffer (1M NaCl, PBS pH 7.5), and urea buffer (2M urea, 50 mM ammonium bicarbonate). Samples were reduced on-bead with 5 mM TCEP at 55°C for 30 min and alkylated with 10 mM iodoacetamide at room temperature for 30 min. To release bound proteins, proteins were subjected to on-bead digestion using 20 µg sequencing grade trypsin (Promega) at room temperature overnight. After overnight digestion, the beads were extensively washed with RIPA buffer, high salt buffer (1M NaCl, PBS pH 7.5), and urea buffer (2M urea, 50 mM ammonium bicarbonate). To release the remaining trypsin digested N-glycosylated peptides from the neutravidin beads, a second on-bead digestion was performed using 2500U of PNGase F (NEB) at 37°C overnight. The eluted fraction was collected using a spin column and then desalted using ZipTips with 0.6 µL C$_{18}$ resin (Millipore Sigma) following standard protocol. Desalted peptides were dried and dissolved in mass spectrometry buffer (0.1% formic acid +2% acetonitrile) prior to LC-MS/MS analysis.

### Mass spectrometry analysis

1 µg of peptide was injected into a pre-packed 0.075 mm x 150 mm Acclaim Pepmap C18 LC column (2 µm pore size, Thermo Fisher) attached to a Q Exactive Plus (Thermo Fisher) mass spectrometer. Peptides were separated using a linear gradient of 3–35% solvent B (Solvent A: 0.1% formic acid, Solvent B: 80% acetonitrile, 0.1% formic acid) over 120 min at 300 µl/min. Data were collected in data-dependent acquisition mode using a top 20 method with a dynamic exclusion of 35 s and a charge exclusion restricted to charges of 2, 3, or 4. Full (MS1) scan spectrums were collected as profile data with a resolution of 140,000 (at 200 *m/z*), AGC target of 3E6, maximum injection time of 120 ms, and scan range of 400–1800 *m/z*. Fragment ion (MS2) scans were collected as centroid data with a resolution of 17,500 (at 200 *m/z*), AGC target of 5E4, maximum injection time of 60 ms with a normalized collision energy at 27, and an isolation window of 1.5 *m/z* with an isolation offset of 0.5 *m/z*.

## Proteomics data analysis

Peptide search and MS1 peak area quantification were performed using ProteinProspector (v.5.13.2) against 20203 human proteins (Swiss-prot database, obtained March 5, 2015) with a false discovery rate of 1%.

Quantitative data analysis was performed using a customized pipeline developed using R (programming language) in RStudio. To ensure stringent quantification of the surface proteome, several filters were applied to eliminate peptide identifications that we believe are due to contamination in the enrichment protocol. Only those peptides containing a PNGase F generated Asn to Asp post-translational modification were considered in the analysis, and proteins with only one quantified peptide were triaged. Proteins not included in a manually curated list of surfaceome proteins (5973 proteins) were also excluded. SILAC ratios were manually calculated from MS1 peak areas and then converted to $\log_2$ ratios normalized to a mean $\log_2$ SILAC ratio = 0 for each dataset. Median log2 SILAC ratios were calculated for each protein and p-values were calculated by a Mann-Whitney test assuming a null hypothesis of $\log_2$(SILAC Ratio)=0.

## GO analysis

Gene sets enriched among proteomic hit genes (p-value<0.01 and absolute $\log_2$ FC >1.75) were determined using David (Huang and Lempicki) using the Homo sapiens gene list as background.

## RNAseq

Total RNA was extracted using the Quiagen RNAeasy kit and RNA amplified using the TruSeq Stranded RNA library creation kit (Illumina) and sequenced to a depth of >20M reads per sample in duplicate. FPKM and fold changes were determined using cufflinks and tophat using hg19 gene annotations.

## Vector design and construction

The vector used to express cell-surface protein ECDs (*Figure 2A*) was generated by Gibson cloning and adapted from the commercially available pFUSE-hIgG1-Fc (InvivoGen) vector. Each ECD was subcloned between an N-terminal IL2 Signal Sequence and a C-terminal TEV cleavage site.

BirA was subcloned into a pLX302 lentiviral vector with a C-terminal KDEL ER localization tag.

We used a previously described vector for expression of Fabs (*Hornsby et al., 2015*). The pFUSE-hIgG1-Fc (InvivoGen) vector was used for expression of IgGs wherein the heavy chain was genetically fused to the hIgG1-Fc and the light chain was expressed on a separate copy of the vector with a C-terminal FLAG tag. The pFUSE-hIgG1-Fc (InvivoGen) vector was also used for expression of the αCD19- αCDCP1. The αCD19 was C-terminally fused to the light chain of the αCDCP1 Fab and the heavy chain was expressed on a separate copy of the vector.

Previously described vectors were used for CRISPRi experiments. Individual sgRNAs were cloned into a pU6 lentiviral vector (Adgene: 46914), dCas9-BFP-KRAB was expressed from a pHR-SFFV lentiviral vector (Adgene: 46911), and the sgRNA library was cloned into a pSICO lentiviral vector (Adgene: 84832).

## Expression of Fc-fusion antigens

HEK293T cells stably expressing ER-localized BirA were grown to ~80% confluency and transiently transfected with 60 μg of ECD-Fc plasmid using 200 μg Linear Poly(ethyleneimine) 25,000 Da (Polysciences). After 6 hr, medium was replaced with DMEM supplemented with 10% Ultra Low IgG FBS (Thermo Fisher), 50 μM biotin, and 1X Pen/Strep. Medium was harvested every 3 days for 6 days and protein was purified by Protein A affinity chromatography and assessed for quality, biotinylation, and integrity by SDS-PAGE.

## Phage display selections

All phage selections were done according to previously established protocols. Briefly, selections with antibody phage Library F (*Persson et al., 2013*) were performed using biotinylated Fc-fusion antigens captured with streptavidin-coated magnetic beads (Promega). Prior to each selection, the phage pool was incubated with 1 μM of biotinylated Fc-domain immobilized on streptavidin beads in order to deplete the library of any binders to the beads or Fc-tag. In total, four rounds of selection

were performed with decreasing amounts of ECD-Fc-fusion antigens (100 nM, 50 nM, 10 nM and 10 nM). To reduce the deleterious effects of nonspecific binding phage, we employed a 'catch and release' strategy, where ECD-Fc-fusion binding Fab-phage were selectively eluted from the magnetic beads by the addition of 2 µg/mL TEV protease. Individual phage clones from the fourth round of selection were analyzed for binding by ELISA.

## Phage ELISAs

ELISAs were performed according to standard protocols. Briefly, 96-well Maxisorp plates were coated with NeutrAvidin (10 µg/ml) overnight at 4°C and subsequently blocked with BSA (2% w/v) for 1 hr at 20 °C. 20 nM of biotinylated ECD-Fc-fusion or Fc-domain was captured on the NeutrAvidin-coated wells for 30 min followed by the addition phage supernatents diluted 1:5 in ELISA buffer (PBS, pH 7.4, 0.05% Tween-20, 0.2% BSA) for 30 min. The bound phage were then detected using a horseradish peroxidase (HRP)-conjugated anti-phage monoclonal antibody (GE Lifesciences 27-9421-01).

## Expression of Fabs

Fabs were expressed according to a previously described protocol. Briefly, C43 (DE3) Pro +*E. coli* containing expression plasmids were grown in TB at 37 °C to an $OD_{600}$ of 0.6–0.8 and then Fab expression was induced by the addition of 1 mM IPTG. Incubation temperature was subsequently reduced to 30 °C and the cultures were allowed to shake for 16–18 hr. Cells were harvested by centrifugation and Fabs were purified by Protein A affinity chromatography. Fab purity and integrity was assessed by SDS-PAGE and intact protein mass spectrometry using a Xevo G2-XS Mass Spectrometer (Waters).

## Flow cytometry

Cells were lifted with Versene (0.04% EDTA, PBS pH 7.4 Mg/Ca free), washed once with PBS pH 7.4, and subsequently blocked with flow cytometry buffer (PBS, pH 7.4, 3% BSA). Fabs (10 µg/mL) or commercial antibodies were added to cells for 30 min at room temperature. Antibodies were detected with addition of Protein A – Alexafluor-647 conjugate (Life Technologies; 1:1000). Cells were extensively washed and fluorescence was quantified using a FACSCanto II (BD Biosciences). All flow cytometry data analysis was performed using FlowJo software and plots were generated using Prism software (GraphPad).

## Immunofluorescence

MCF10A KRAS and empty vector cells were plated on glass bottom imaging plates (MatTek) and allowed to incubate for 24 hr at 37 °C under 5% $CO_2$. Cells were fixed with 1% paraformaldehyde for 10 min, followed by addition of Fabs (10 µg/mL) in imaging buffer (PBS, pH 7.4, 3% BSA) for 30 min. Fab binding was detected by the addition of a Protein A – Alexafluor-546 conjugate (Life Technologies; 1:1000) in imaging buffer containing Hoescht blue (2 µg/mL). Cells were imaged on a Zeiss Observer Z1 microscope using oil immersion. All immunofluorescence figures were generated using unbiased post-processing in Photoshop (Adobe).

## Surfaceome sgRNA library specifications

Surfaceome genes were selected as a subset of the UniProt Human Proteome (19,942 genes). Genes were filtered based on UniProt subcellular location annotation to remove those proteins not annotated as localized to the 'Cell Membrane' (2015 genes), 'Membrane' (2162 genes), or 'Extracellular Space' (280 genes). The gene list was further filtered to remove those genes not predicted by UniProt to contain a signal sequence (1605 genes). Finally, a set of RAS pathway genes (21 genes) was added (1626 genes). sgRNAs were designed using a previously described algorithm (Horlbeck and Weissman) and were synthesized in batch by Agilent. 150 non-targeting sgRNAs were included as negative controls.

## CRISPRi library cloning

CRISPRi library sgRNA sequences were designed using previously described methods (*Gilbert et al., 2014*; *Horlbeck et al., 2016*). Oligonucleotide pools were synthesized by Agilent.

Each library was amplified by PCR, digested with BstXI and BlpI, and cloned into an sgRNA expression vector.

## Preparation of CRISPRi lentiviral library

To produce virus, HEK293T cells were transfected with a mixture of library plasmid DNA and second-generation lentiviral packaging plasmids at ~80% confluence. Transfection of the plasmids was performed using TransIT-LTI Transfection Reagent (Mirus, MIR 2306). Medium was changed to complete DMEM after 6 hr of incubation with transfection mixture. The supernatant was harvested and cleared by passing through a 0.45 µm PVDF filter 72 hr post transfection and immediately used for infection.

## CRISPRi screen

CRISPRi MCF10A cell lines were infected with the surfaceome sgRNA library as previously described (*Gilbert et al., 2014*) (*Kampmann et al., 2013*). The infection was scaled to achieve a multiplicity of infection of less than one sgRNA per cell. Two days after infection, cells were treated with 2 µg/ml puromycin (Sigma Aldrich) for 72 hr to select for sgRNA expression, and then grown in fresh medium without puromycin for 48 hr recovery. MCF10A empty vector cells were grown in standard MCF10A medium whereas the MCF10A KRAS$^{G12V}$ cells were either grown in standard MCF10A medium or MCF10A medium lacking insulin and epidermal growth factor (EGF). Cells were grown for 12 days, and at least 10 population doublings. Cells were maintained at a confluency between 20–80%, and at numbers continually maintaining a library coverage of at least 1000 cells per sgRNA. Populations of cells expressing this library of sgRNAs were harvested at the outset of the experiment (T0) and after 12 days (T12). Genomic DNA was harvested from all samples; the sgRNA-encoding regions were then amplified by PCR and sequenced on an Illumina HiSeq-2500 using custom primers with previously described protocols at high coverage (*Gilbert et al., 2013*). Two biological replicates of each screening condition were performed. From this data, we quantified the frequencies of cells expressing different sgRNAs in each sample and quantified the phenotype of each sgRNA as the average phenotype of the three strongest sgRNAs.

## Assessment of CRISPRi phenotypes for individual sgRNAs

Individual phenotype validation experiments for sgRNAs from the CRISPRi screens, were performed as competitive growth experiments with a mixed culture of sgRNA-GFP infected and non-infected cell populations of MCF10A empty vector and KRAS$^{G12V}$. Following infection, cells were selected in puromycin (2 µg/mL) for 3 days and then allowed to recover for 2 days. Infected and non-infected cells were counted and seeded at a 1:1 ratio in 96 well plates. In parallel, infected cells were grown in mono-culture so that sgRNA expression could be monitored. Triplicate samples for each sgRNA were grown under standard conditions for the MCF10A empty vector, or with media lacking insulin and EGF for the MCF10A KRAS$^{G12V}$. All cells grown in mono-culture were grown under standard conditions. The percentage of cells expressing GFP (indicating sgRNA expression) was measured at several time points (0 days, 5 days, 12 days). The relative fraction of sgRNA positive cells was determined by the percentage of cells that maintained expression of GFP (indicating sgRNA expression) normalized to the percentage of GFP positive cells on the day of doxycycline addition (day 0).

## Expression of BiTE and IgG

Expi293 (Life Technologies) cells were transiently co-transfected with two pFUSE (InvivoGen) vectors harboring either the Fab heavy chain and the Fab light chain genetically fused to the αCD19 scFv, or the Fab heavy chain fused to a human Fc and the Fab light chain, at a ratio of 1:1 for BiTE and IgG respectfully. The ExpiFectamine 293 transfection kit (Life Technologies) was used for transfections as per manufacturer's instructions. Cells were incubated for 7 days at 37 ˚C in a 5% CO$_2$ environment before the supernatants were harvested by centrifugation. Protein was purified by Protein A affinity chromatography and assessed for quality and integrity by SDS-PAGE.

## Antibody drug conjugate cell killing assay

Antibody drug conjugate cell killing assays were performed using a Fab Anti-Human IgG Fc-MMAF Antibody with Cleavable Linker (Moradec) following manufacture's protocol. Briefly, 5000 HPAC or

HPNE cells were plated in each well of a 96-well plate, and IgG (4 nM initial, 5-fold serial-dilutions, six times) or Staurosporine (2 µM) was added. After 10 min incubation, 40 nM of the Fab anti-human IgG Fc-MMAF antibody with cleavable linker was added to a final volume of 100 µL. Cells were then allowed to incubate for 72 hr at 37°C under 5% $CO_2$. After the incubation period, 70 µL of CellTiter-Glo Reagent (Promega) was added to each well followed by shaking incubation at room temperature for 10 min. Luminescence was then quantified using an Infinite M200 PRO plate reader (Tecan).

## BiTE assay

Jurkat NFAT-reporter cells were mixed with HPAC or HPNE target cells at a ratio of 1:1. Bispecific T-cell engaging antibody (BiTEs) (anti-CDCP1-antiCD19) or Fab (anti-CDCP1) was diluted in medium and added to cell mixtures at a final concentration of 1 nM. After 20 hr incubation at 37°C, cells were pelleted by centrifugation. NFAT-dependent GFP reporter expression was quantified by flow cytometry using a FACSCanto II (BD Biosciences).

## Immunoblotting

Cells were plated at approximately $0.5 \times 10^6$ cells/well in a 6-well plate and cultured overnight before drug treatment. Medium was replaced with medium supplemented with Gefitinib (2 µM), SB590885 (2 µM), PD0325901 (2 µM), XL147 (2 µM), MK2206 (2 µM), or Vehicle (0.2% DMSO). The cells were further incubated at 37°C for 48 hr, after which the cells were washed with PBS and lysed with RIPA Lysis and Extraction Buffer (Thermo Scientific) supplemented 1X Protease Inhibitor Cocktail (Sigma) at 4°C for 30 min. Immunoblotting was performed using AKT(pan) (Cell Signaling mouse mAB, #2920), Phospho-AKT (Thr308) (Cell Signaling rabbit mAB, #2965), p44/42 MAPK (Erk1/2) (Cell Signaling rabbit mAB, #4695), Phospho-p44/42 MAPK (Erk1/2) (Thr202/Tyr204) (Cell Signaling rabbit mAB, #4370), CDCP1 (Cell Signaling rabbit mAB, #13794), Tubulin (Sigma mouse mAB #T6199), IRDye 680RD Goat anti-Mouse (Licor # 925–68070), and IRDye 800CW Donkey anti-Rabbit (Licor # 925–32213) antibodies.

## Coupling of the desferrioxamine chelate on the antibody (Fab)

110 µL of Fab (anti-CDCP1) at a concentration of 4.2 mg/mL was dispersed in 100 µL of 0.1 M sodium bicarbonate buffer (pH 9.0). The pH was adjusted to 9.0 and the final reaction mixture was adjusted to a total volume of 0.5 mL by adding a sufficient amount of 0.1 M sodium bicarbonate buffer. Df-Bz-NCS (p-isothiocyanatobenzyl-desferrioxamine) was dissolved in DMSO at a concentration of 10 mM. Df-Bz-NCS solution was added to the antibody solution to give a three molar excess of the chelator over the molar amount of Fab. The Df-Bz-NCS was added in steps of 2 µL and mix rigorously during the addition. The concentration of DMSO was kept below 2% of the total reaction mixture in order to avoid any precipitation. After 30 min at 37°C, the reaction mixture was purified via a PD-10 column pre equilibrated by 20 mL of gentisic acid solution (5 mg/mL of gentisic acid in 0.25 M sodium acetate (pH 5.4–5.6)). The Fab-DFO solution was eluated in gentisic acid solution, and the pH was adjusted to seven by addition of NaOH (1 M). Then, the solution was aliquoted and stored at −20°C until the day of radiolabeling.

## $^{89}$Zr Radiochemistry

[$^{89}$Zr]Zr-oxalic acid solution (5mCi; 10 µl) was neutralized with 2 M $Na_2CO_3$ (5 µL). After 3 min, 0.30 mL of 0.5 M HEPES (pH 7.1–7.3) and 0.5 mg of DFO-Fab (pH 7) were added into the reaction vial. After incubation (120 min) at 37°C, the radiolabeling efficiency was determined by ITLC using chromatography strips and 20 mM citric acid (pH 4.9–5.1). The radiolabeling efficiency was consistently >98.5%.

## $^{125}$I Radiochemistry

Iodination with iodine-125 was done in pre-coated iodination tubes (Pierce). 100 µg of anti-CDCP1 Fab was dispersed in 100 µL of PBS solution and added to the pre-coated iodination tubes. Separately in a 1.5 mL eppendorf tube, 1 µL of HCl (0.2 M), 2.5 µL of phosphate buffer (0.5 M, pH = 8), 10 µL of potassium iodide solution (1 mg/ml) was prepared. 1 mCi of iodine-125 was added into the tubes and the previous solution was then mixed in the iodination tubes. After 15 min of reaction the

solution was purified via PD10 column pre equilibrated with 20 mL of PBS solution. The purity was assessed via iTLC, and $^{125}$I-FAB was consistently >98% pure.

## Cell culture

The MCF10A cells were maintained in DMEM medium, supplemented with 5% Horse Serum (Invitrogen), 20 ng/mL EGF (Invitrogen), 500 ng/mL Hydrocortisone (Sigma), 100 ng/mL Cholera Toxin (Sigma), 10 µg/mL Insulin (Sigma), and 1X Pen/Strep. The SW620, SW403, SW1116, and CAPAN-1 cell lines were a gift form the laboratory of Frank McCormick and were maintained in McCoy's 5A medium, supplemented with 10% FBS and 1X Pen/Strep. The Panc1, BXPC3, CAPAN-2, PL5, H5667, HPAC, MiaPaCa-2, and HPNE cells were a gift from the laboratory of E. Scott Seeley and were maintained in IMDM supplemented with 10% FBS and 1X Pen/Strep. The H1792, H358, and H1299 cells were a gift from the laboratory of Kevan Shokat and were maintained in RPMI supplemented with 10% FBS and 1X Pen/Strep. The HEK293T cell lines were cultured in DMEM supplemented with 10% FBS and 1X Pen/Strep. Jurkat NFAT reporter cell lines were cultured in RPMI supplemented with 10% FBS, 2 mg/mL G418 (Thermo) and 1X Pen/Strep. Cell line identities were authenticated by morphological inspection. Symptoms for mycoplasma contamination were not observed and thus no test for mycoplasma contamination was performed. To the best of our knowledge, all cell lines that were received as gifts were previously authenticated and tested for mycoplasma.

To construct the CRISPRi cell lines, MCF10A empty vector and KRAS$^{G12V}$ cells were lentivirally transduced with constructs encoding dCas9-BFP-KRAB. Three rounds of sorting for stable BFP expression by flow cytometry using a BD FACS Aria2 were used to generate homogeneous polyclonal populations of each CRISPRi cell line. A second round of lentiviral transduction was used to introduce expression of sgRNAs. Cells expressing sgRNAs were selected for in puromycin (2 µg/mL) and validated by confirming either GFP (single sgRNA) or BFP (sgRNA library) expression using fluorescence microscopy or flow cytometry.

To construct the ECD displaying HEK293 cell lines, Flp-In T-REx (ThermoFisher) cells were co-transfected with the pOG44 vector (ThermoFisher) and a construct encoding each ECD fused to the transmembrane anchoring domain of platlet-erived Growth Factor with a YPET tag in the pcDNA5/FRT Mammalian Expression vector (ThermoFisher). Cells displaying each ECD were selected for in DMEM supplemented with 10% FBS and 50 µg/mL Hygromycin (ThermoFisher). Surface display was confirmed by fluorescence microscopy detection of the YPET tag.

## Saturation binding assay

The number of receptors per cell was determined by a saturation binding experiment performed in A549, SW62D, MiaPAca2, and HPAC cells using $^{125}$I-CDCP1 Fab. 60,000 cells were plated into wells of 12 well plates. The saturation binding assays were done in a range of 9 different concentrations between 0.65 to 80 nM of $^{125}$I-CDCP1 Fab in each well. The non-specific binding was determined at three different concentrations (0.65, 20 and 80 nM) by co-incubation of cells with 1000 fold excess over the Kd of the unlabeled anti-CDCP1 Fab. Following incubation for 1 hr, the cells were washed twice with ice cold PBS and retained for analysis (unbound fraction). The cells were lysed with 1 mL of 1M NaOH and collected (cell associated fraction). The unbound and cell-associated fractions were counted in a gamma counter and expressed as a percentage of the total activity added per number of cells. Experiments were performed in triplicate. The specific binding was obtained by subtracting the non-specific binding from the total binding. A Rosenthal plot was used to determine the Bmax.

## Immunoreactivity fraction

The immunoreactive fraction was determined by measuring the percentage of radiolabeled Fab bound to antigen under conditions of antigen excess. Using a concentrated cell suspension of HPAC cell lines, five different dilutions of cells ranging from 0.5 to 10 million cells/mL were prepared in PBS solution containing BSA. For each cell concentration, $^{89}$Zr-DFO-CDCP1 (10 µCi) was incubated for 1 hr at room temperature. Following the incubation, the cell suspensions were washed twice with ice-cold PBS and centrifuged. The unbound radioactivity was removed and counted along with the cell-bound activity. The inverse of cell-bound radioactivity (total over bound) was plotted against the inverse of cell concentration. The data were fitted with a least squares linear regression method

(PRISM). The Y-intercept of the regression line represents the inverse of the immunoreactive fraction.

## In vivo imaging

Three to five week old male nu/nu immunocompromised mice were purchased from Charles River. Nu/Nu mice were inoculated with $1 \times 10^6$ HPAC or A549 cells subcutaneously into one flank in a 1:1 mixture (v/v) of PBS and Matrigel (Corning). Tumors were palpable within 21–30 days with HPAC and 14–21 days with A549 after injection. Tumor-bearing mice (n = 5 per treatment arm) received between 50 to 300 µCi of $^{89}$Zr-CDCP1 or $^{89}$Zr-heat denatured Fab solution in 100 µL saline solution volume intravenously using a custom mouse tail vein catheter with a 28-gauge needle and a 100–150 mm long polyethylene microtubing. ~300 µCi was injected for the mice for imaging and ~50 µCi for the mice for biodistribution. The mice were imaged at different time points (4 hr, 8 hr, 24 hr) on a dedicated small animal PET/CT scanner (Inveon, Siemens Healthcare, Malvern, PA). Animals were scanned for 20 min for PET, and the CT acquisition was 10 min. The coregistration between PET and CT images was obtained using the rigid transformation matrix from the manufacturer-provided scanner calibration procedure since the geometry between PET and CT remained constant for each of PET/CT scans using the combined PET/CT scanner. Animals were anesthetized with gas isoflurane at 2% concentration mixed with medical grade oxygen. The photon attenuation correction was performed for PET reconstruction using the coregistered CT-based attenuation map to ensure the quantitative accuracy of the reconstructed PET data.

## Drug treatment for in vivo experiments

Tumor-bearing mice were treated with MEK inhibitor (PD0325901). PD0325901 (Sigma) was suspended in HPMT solution (0.5% w/v hydroxypropyl-methylcellulose dissolved in water plus 0.2% v/v Tween 80). Tumor-bearing mice were treated once daily via oral gavage with PD0325901 (25 mg/kg/d) for 6 days (4 days before the injection and 2 days following the radiotracer injection).

## Biodistribution studies

Biodistribution studies were conducted to evaluate the uptake of $^{89}$Zr-CDCP1 in mice bearing subcutaneous tumors. At a dedicated time after radiotracer injection, animals were euthanized by $CO_2(g)$ asphyxiation, and 14 tissues (including the tumor) were removed, weighed and counted on a gamma-counter for accumulation of $^{89}$Zr-radioactivity. The mass of $^{89}$Zr-CDCP1 formulation injected into each animal was measured and used to determine the total number of counts (counts per minute, [c.p.m.]) by comparison to a standard syringe of known activity and mass. Count data were background- and decay-corrected and the tissue uptake measured in units of percentage-injected dose per gram (%ID/g) was calculated by normalization to the total amount of activity injected.

## Acknowledgements

We thank E Scott Seeley, Martin McMahon, Charles Craik, Frank McCormick, Brian Lee, Tet Matsuguchi, Jia Liu, Karolina Wypisniak, Michael Hornsby, Jacqueline Villalta, Yun Mou, James Koerber, and Victoria Assimon for helpful discussions and technical advice. We thank Sachdev Sidhu (University of Toronto) for providing the phage-displayed Fab library. We thank Sourav Bandyopadhyay (UCSF) and Maggie Martins (UCSF) for generating and providing the MCF10A empty vector and KRAS$^{G12V}$ cell lines. We thank Frank McCormick and Jacqeline Galeas for providing the SW620, SW403, SW1116, and CAPAN-1 cell lines. We thank E Scott Seeley (Salvo LLC) for providing the Panc1, 3XPC3, CAPAN-2, PL5, H5667, HPAC, and HPNE cell lines. We thank Kevan Shokat for providing the H1792, H358, and H1299 cell lines. We thank Arthur Weiss (UCSF) and Theresa Kadlecek (UCSF) for providing the NFAT-dependent GFP reporter Jurkat cell line. Funding was provided by R01 grants from the NIH (CA191018 and GM097316). AJM was supported by a predoctoral fellowship from the NSF GRFP.

# Additional information

## Competing interests

James A Wells: Celgene Corp provided some of the funding for this work. The other authors declare that no competing interests exist.

## Funding

| Funder | Grant reference number | Author |
| --- | --- | --- |
| National Institutes of Health | CA191018 | Alexander J Martinko<br>Olivier Julien<br>James A Wells |
| Celgene | | Alexander J Martinko<br>James A Wells |
| National Institutes of Health | GM097316 | Alexander J Martinko<br>Olivier Julien<br>James A Wells |
| National Science Foundation | Graduate Research Fellowships Program | Alexander J Martinko |

The funders had no role in study design, data collection and interpretation, or the decision to submit the work for publication.

## Author contributions

Alexander J Martinko, Conceptualization, Data curation, Formal analysis, Investigation, Methodology, Writing—original draft, Writing—review and editing; Charles Truillet, Olivier Julien, Formal analysis, Investigation, Writing—review and editing; Juan E Diaz, Conceptualization, Formal analysis; Max A Horlbeck, Resources, Formal analysis, Methodology; Gordon Whiteley, Conceptualization, Resources, Writing—review and editing; Josip Blonder, Conceptualization, Resources; Jonathan S Weissman, Conceptualization, Resources, Methodology; Sourav Bandyopadhyay, Michael J Evans, James A Wells, Conceptualization, Resources, Investigation, Writing—review and editing

## Author ORCIDs

Alexander J Martinko (iD) http://orcid.org/0000-0002-4025-959X

Olivier Julien (iD) http://orcid.org/0000-0001-7068-7299

Max A Horlbeck (iD) http://orcid.org/0000-0002-3875-871X

Jonathan S Weissman (iD) http://orcid.org/0000-0003-2445-670X

Sourav Bandyopadhyay (iD) http://orcid.org/0000-0003-3120-923X

James A Wells (iD) http://orcid.org/0000-0001-8267-5519

## Ethics

Animal experimentation: This study was performed in strict accordance with the recommendations in the Guide for the Care and Use of Laboratory Animals of the National Institutes of Health. All of the animals were handled according to approved institutional animal care and use committee (IACUC) protocols (AN110806-03C) of the University of California San Francisco. Animals that were studied with PET/CT were anesthitized with a mixture of isoflurane/oxygen and humanely euthanized with $CO_2(g)$.

## Decision letter and Author response

Decision letter https://doi.org/10.7554/eLife.31098.sa1
Author response https://doi.org/10.7554/eLife.31098.sa2

## Additional files

**Supplementary files**

• Supplementary file 1. SILAC surface proteomics dataset for MCF10A KRAS[G12V] compared to MCF10A Empty Vector.

• Supplementary file 2. SILAC surface proteomics dataset for MCF10A KRAS[G12V] cells treated with MEKi or Vehicle.

• Supplementary file 3. SILAC surface proteomics dataset for MCF10A KRAS[G12V] cells treated with AKTi or Vehicle.

• Supplementary file 4. SILAC surface proteomics dataset for MCF10A KRAS[G12V] cells treated with EGFRi or Vehicle.

• Supplementary file 5. RNAseq dataset for MCF10A KRAS[G12V] compared to MCF10A Empty Vector.

• Supplementary file 6. Sequences of all sgRNAs contained in the surfaceome CRISPRi library.

• Supplementary file 7. CRISPRi growth screen datasets for MCF10A KRAS[G12V] (Minus Growth Factors), MCF10A KRAS[G12V] (Plus Growth Factors), and MCF10A Empty Vector (Plus Growth Factors).

• Transparent reporting form

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
