## [Decision Letter]

Thank you for submitting your article "Targeting RAS Driven Cancers with Antibodies to Upregulated and Functionally Important Cell-Surface Proteins" for consideration by *eLife*. Your article has been favorably evaluated by Charles Sawyers (Senior Editor) and three reviewers, one of whom is a member of our Board of Reviewing Editors. The reviewers have opted to remain anonymous.

The reviewers have discussed the reviews with one another and the Reviewing Editor has drafted this decision to help you prepare a revised submission.

Summary:

Martinko and co-authors have generated a toolkit of recombinant antibodies targeting cell surface proteins induced by activated Ras in cancer cells as a therapeutic strategy. Initially using an isogenic cell line pair based on MCF10A mammary epithelial cells with or without mutationally activated KRAS, they characterized the cell surface proteome by mass spec to identify Ras-induced surface proteins. These were enriched for proteins involved in cell adhesion and migration, and the effects were largely dependent on Ras-MEK pathway signaling. Parallel analysis of RNA expression revealed only a small correlation between Ras-induced surface proteins and corresponding RNAs, suggesting that the mass spec approach can reveal novel candidate targets. They made recombinant antibodies targeting some of these surface proteins, which validated the observed Ras-induced surface expression. They then used CRISPi screening of 1600 annotated surface proteins to look for those demonstrating functional dependency specifically in Ras-transformed cells. They prioritized efforts focused on one of these genes, which encodes CDCP1, an integrin-associated protein previously implicated in cancer/ Using CDCP1 antibodies conjugated to a cytotoxin, they demonstrated the potential utility of such an antibody-drug conjugate targeting CDCP1 as a therapeutic.

Overall, while these studies are technically well executed, and represent a potentially useful application of recombinant antibody technology coupled with mass spec analysis of the surface proteome of cancer cells, the manuscript reads like a collection of somewhat orthogonal observations that are not logically well-connected. Moreover, as described below, the emphasis on CDCP1 detracts from the overall enthusiasm, since this has been previously implicated as a target in Ras-driven cancers.

Essential revisions:

1) The logical flow, as presented, complicates the interpretation of the key conclusions. For example, the authors emphasize that the correlation was weak between cell surface proteome and RNA expression; however, the highlighted example CDCP1, has already been reported to be a Ras-induced gene at the RNA level. Similarly, the authors used CRISPRi to establish a list of surface proteins that are functionally required in Ras-transformed cells; however, they then went on to use the CDCP1 antibody to deliver a toxic payload to cancer cells, rather than exploring a function-blocking antibody. This is confusing since it raises a question about the rationale for the CRISPRi analysis.

2) CDCP1 is not an ideal candidate to emphasize as the key example, largely due to the fact that it has been previously published as a Ras-induced candidate target. Notably, in the Discussion, the authors state that,"our work now demonstrates MAPK dependent expression of this protein […]"; however, the previous report from Uekita (which is cited) had already demonstrated the role of MAPK signaling in Ras-induced CDCP1 expression. So, the novelty is quite limited here.

3) The authors argue strongly and for the most part convincingly that the use of MCF10A was reasonable for these experiments. However, the authors should discuss that they may well have missed important cell surface targets using this approach. This is particularly salient given the wide variability of expression of the proteins that they found across other cell lines. MCF-10A is a curious choice for these studies when considering that breast cancer is often highlighted as one of the few cancer types that never harbor KRAS mutations. The authors should probably highlight in more detail why they chose this cell line as the starting point, even if it is just for technical reasons. There are now multiple examples (from different tissue types) of cancer cell lines that are engineered to be +/- KRAS.

4) To address the points raised above, the authors should substantially revise the manuscript to emphasize the recombinant antibody technology as applied to targeting the surface proteome of cancer cells, while reducing the significance of selectively targeting Ras mutant cancer cells. Furthermore, the manuscript should include a more realistic discussion of the implications of the work. Not all KRAS mutant cancers are driven by MAPK and not all of these will have CDCP1 up-regulated. Moreover, since no normal cells or tissues were analyzed (neither MCF-10A or HPNE cells are normal), it is unclear from these experiments whether a therapeutic window really exists for CDCP1 targeting, especially given that no anti-tumor activity is shown.

5) For Figure 3C, the authors mentioned that the 8 cell lines were selected because pancreatic, lung, and colorectal cancers have the highest frequency of KRAS mutation. It is very nice to see that CDCP1 showed high surface expression in almost every cell line tested. However, the other 6 proteins did not. Although a full characterization of all of these proteins goes beyond what is necessary in this report, it would be helpful to have a more robust discussion of this issue.

6) In addition to Figure 5C, it would be more convincing to repeat the same experiment in other cell lines: pancreatic or other lineages that have been shown in Figure 3C to have high expression of CDCP1.

7) In Figure 6B, the BRAF inhibitor SB590885 did not decrease phospho-ERK levels? Is this correct?

8) In Figure 6B, is it a robust result that the AKT inhibitor MK2206 only decrease the surface localization of CDCP1 but not CDCP1 total protein level?

9) NCI-H1299 cells have wild-type KRAS and NRAS-Q61K mutation, not KRAS-Q61 as stated in the text and Figure 6C. More RAS-mutant cell lines should be tested to further support the claim that CDCP1 expression is coupled with activation of MAPK signaling.

10) The authors have used publicly available databases to assess whether CDCP1 is overexpressed in KRAS mutant cancers. It would be helpful to determine this by a different method than probing gene expression (mRNA) as this is not entirely reliable.

11) The experiment described in Figure 3C is not very informative given that no RAS WT cell lines are shown. Moreover, without some sort of KRAS perturbation (knockdown or MEK inhibition), there is no evidence that the expression of any of these proteins has anything to do with RAS mutation status.

12) The type of data shown in Figure 5Ais not very informative from a therapeutic point of view. Even though KRAS mutant and WT are statistically different at the population level, in truth only 7% of PDAC over-express CDCP1. 91% of PDAC (in the TGCA dataset) have mutant KRAS, so, regardless of the claims made in the paper, the general relationship between mutant KRAS is unclear. The Discussion should probably be more frank about this. Moreover, Figure 1F shows a less than impressive correlation between transcriptomics and proteomics, yet they are trying to make an important conclusion about CDCP1 protein expression from transcriptomics data on primary tumors. It would be useful if they could comment specifically on the correlation between RNA and protein for CDCP1 in their original analysis.

[Editors' note: further revisions were requested prior to acceptance, as described below.]

Thank you for resubmitting your work entitled "Targeting RAS Driven Human Cancer Cells with Antibodies to Upregulated and Essential Cell-Surface Proteins" for further consideration at *eLife*. Your revised article has been favorably evaluated by Charles Sawyers (Senior Editor) and a Reviewing Editor.

The manuscript has been improved but there are some remaining issues that need to be addressed before acceptance, as outlined below:

The Abstract, which emphasizes biological findings, remains unchanged. For example, the statement, "we discovered a signature of proteins critical for metastasis that are upregulated on cells transformed with KRASG12V, and driven by MAPK pathway signaling." is misleading in light of the findings presented. Similarly, the phrasing of the statement describing CDCP1 findings still leads readers to conclude that this protein has been newly identified by these studies. The authors need to edit the Abstract to appropriately represent the key novel claims, and the emphasis on the new technology.

The authors have not adequately addressed a concern that was raised regarding the transition from the CRISPRi discovery of a requirement for CDCP1 and the experiments in which it was targeted using an ADC strategy (subsections “Functional characterization of the KRAS surfaceome in MCF10As using a CRISPRi screen” and “Antibodies can selectively deliver toxic and immunotherapy payloads to mutant KRAS pancreatic cancer cells”). In the reply to reviewer comments, the authors refer to a statement they make where they write: "One of the most common mechanisms of resistance to targeted therapies is loss of expression of the protein target." But this is not true. In fact, most targeted therapies are directed to proteins that are required by cancer cells and such targets are therefore not typically reduced in expression as a resistance mechanism. The authors should remove or restate this. More notably, the authors should explain in the transition between these two sections that the dependency of RAS-transformed cancer cells on CDCP1, together with its high surface expression, makes it an attractive ADC target-which is unlikely to be selected against as a resistance mechanism due to its requirement in these cancer cells. Therefore, they explored the ADC approach. They should also comment on the fact that their antibody to CDCP1 is apparently not a function-blocking antibody-since it does not seem to affect cell viability.

---

## [Author Response]

Essential revisions:1) The logical flow, as presented, complicates the interpretation of the key conclusions. For example, the authors emphasize that the correlation was weak between cell surface proteome and RNA expression; however, the highlighted example CDCP1, has already been reported to be a Ras-induced gene at the RNA level.

We appreciate this comment as some may view our work on CDCP1 as redundant. Importantly, we show there is generally a weak correlation between the RNAseq and the surface proteome data.

“The correlation between expression level changes observed in the SILAC proteomic data with those from RNAseq data (Figure 1F) was modest but significant (R^2^ =0.422).”

Thus, the fact that others may have seen the up-regulation of CDCP1 mRNA does not give us confidence that it should be so at the protein level. Raising recombinant antibodies is a significant effort and one that is much riskier based on mRNA data alone. Proteomic and RNAseq data sets considered together provided us greater confidence to embark on antibody discovery efforts. We were also heartened to see that the increase in target expression from the proteomics was confirmed by the antibodies we developed, a completely orthogonal metric.

Similarly, the authors used CRISPRi to establish a list of surface proteins that are functionally required in Ras-transformed cells; however, they then went on to use the CDCP1 antibody to deliver a toxic payload to cancer cells, rather than exploring a function-blocking antibody. This is confusing since it raises a question about the rationale for the CRISPRi analysis.

Functionally blocking antibodies are a small subset of antibodies that can be raised, and frankly, are challenging to discover, since one needs great mechanistic insight into the target, which we currently do not have for CDCP1. On the other hand, knowing that a target is important *and* upregulated in cancer cells makes it very attractive as an ADC or immune- oncology target. In this case one needs a binding antibody (like anti-CD19) and to know that the target is critical so resistance cannot be easily achieved by loss of expression the target protein, which is the most common means of resisting a therapeutic antibody for cancer. Thus we feel that proteomics and CRISPRi functional genomics are important companions to discover new targets. To clarify this point we have edited the text:

“One of the most common mechanisms of resistance to targeted therapies is loss of expression of the protein target. […] Towards identification of such candidates, we explored how oncogenic KRAS signaling can influence the dependence of cells on surface proteins.”

2) CDCP1 is not an ideal candidate to emphasize as the key example, largely due to the fact that it has been previously published as a Ras-induced candidate target. Notably, in the Discussion, the authors state that, "our work now demonstrates MAPK dependent expression of this protein […]"; however, the previous report from Uekita (which is cited) had already demonstrated the role of MAPK signaling in Ras-induced CDCP1 expression. So, the novelty is quite limited here.

The reviewers request that we further address the issue of novelty of CDCP1 expression beyond that reported by Uekita. Uekita only reported this connection in non-small cell lung cancers. We now demonstrate the connection between RAS and CDCP1 in lung, pancreatic, and colorectal cell models for mutant RAS cancers and include this new data in Figure 3D. Moreover the data shows a very broad MEKi dependent decrease for CDCP1 relative to AKTi or EGFRi, and that this decrease is much more pronounced than other targets in the panel.

Because we now demonstrate the connection between RAS signaling and CDCP1 in the context of Figure 3, we chose to eliminate the more sparse data set and diagram from panels A and B in Figure 6.

3) The authors argue strongly and for the most part convincingly that the use of MCF10A was reasonable for these experiments. However, the authors should discuss that they may well have missed important cell surface targets using this approach. This is particularly salient given the wide variability of expression of the proteins that they found across other cell lines. MCF-10A is a curious choice for these studies when considering that breast cancer is often highlighted as one of the few cancer types that never harbor KRAS mutations. The authors should probably highlight in more detail why they chose this cell line as the starting point, even if it is just for technical reasons. There are now multiple examples (from different tissue types) of cancer cell lines that are engineered to be +/- KRAS.

We appreciate this comment and modified the text to further elaborate the advantages and caveats of MCF10A in the text:

“To begin to isolate the effects of KRAS transformation on the cell surfaceome, we chose a well-characterized, non-tumorigenic immortalized epithelial cell line, MCF10A, to generate an isogenic model for KRAS transformation (Debnath et al., 2003; Martins et al., 2015). […] However, to determine the therapeutic potential of exploiting differences in the surfaceome in RAS-driven cancers, we needed to quantitatively measure the extent to which these proteins change.”

4) To address the points raised above, the authors should substantially revise the manuscript to emphasize the recombinant antibody technology as applied to targeting the surface proteome of cancer cells, while reducing the significance of selectively targeting Ras mutant cancer cells.

In response to the reviewers’ suggestions that we emphasize the platform technology over the specific biology, we have now combined Figures 5 and 6 to one figure that is focused on demonstrating proof of concept applications of our antibodies to PDAC cells harboring mutant RAS. The combined figure presents systematic approaches to demonstrate the utility of the antibodies generated from our technological pipeline. These methods are straightforward and can be readily applied to any of the recombinant antibodies we generate, exemplified in this case for CDCP1.

Furthermore, the manuscript should include a more realistic discussion of the implications of the work. Not all KRAS mutant cancers are driven by MAPK and not all of these will have CDCP1 up-regulated. Moreover, since no normal cells or tissues were analyzed (neither MCF-10A or HPNE cells are normal), it is unclear from these experiments whether a therapeutic window really exists for CDCP1 targeting, especially given that no anti-tumor activity is shown.

Our new data in Figure 3 do suggest that CDCP1 expression is broadly under MAPK control but we take the point that not all KRAS mutant cancers are driven by MAPK and that the data we have is for cell lines in culture, not in a tumor or normal tissues. We do show in Figure 5—figure supplement 1B-C that CDCP1 is not broadly expressed at high levels compared to HER2, a validated cancer target, but clearly more work is needed. Thus we have modified the text:

“More extensive expression profiling of human tissues is needed to truly estimate the therapeutic potential of this protein. However, we believe that the observations described herein provide motivation for the further exploration of CDCP1 as a target for RAS mutant cancers.”

5) For Figure 3C, the authors mentioned that the 8 cell lines were selected because pancreatic, lung, and colorectal cancers have the highest frequency of KRAS mutation. It is very nice to see that CDCP1 showed high surface expression in almost every cell line tested. However, the other 6 proteins did not. Although a full characterization of all of these proteins goes beyond what is necessary in this report, it would be helpful to have a more robust discussion of this issue.

We appreciate the enthusiasm expressed for this experiment and we agree that a more thorough discussion of the lack of expression of several of the targets merits discussion, which we have now added below. These comments include discussion of the newly added inhibitor experiment described above in Figure 3:

“Most strikingly we observed that one protein, CDCP1, was ubiquitously abundant in expression across the entire panel. […] Moreover, we believe this collection of highly specific recombinant antibodies constitute an expandable and enriched tool kit that may be useful for interrogating RAS transformed cells and tissues.”

6) In addition to Figure 5C, it would be more convincing to repeat the same experiment in other cell lines: pancreatic or other lineages that have been shown in Figure 3C to have high expression of CDCP1.

We thank the reviewers for this suggestion and have performed this ADC experiment in two additional cell lines to add further credence to our claims about the potential use of our antibodies to deliver toxic payloads to cancer cells. This data has been added as Figure 5—figure supplement 2.

Brief discussion of this new supplemental figure was also added to the text:

“More than 60% reduction of viability was observed at 0.8 nM (Figure 5D), suggesting that the CDCP1-selective antibody can selectively internalize and deliver the cytotoxic payload. Similar levels of toxicity were observed in two additional mutant RAS cancer cell lines (Figure 5—figure supplement 2).”

7) In Figure 6B, the BRAF inhibitor SB590885 did not decrease phospho-ERK levels? Is this correct?

Indeed, we were also surprised that SB590885 did not decrease phospho-ERK levels but instead appeared to slightly increase them. We then became aware of the fact that this is not the appropriate inhibitor for BRAF because it can paradoxically activate RAF signaling in RAS mutant cells as reported in Peng, S.-B., et al. (2015). Cancer Cell, 28(3), 384–398.

“Additionally, a type I RAF inhibitor, SB590885, also induced significant BRAF/CRAF heterodimer and CRAF/CRAF homodimer. However, in contrast with LY3009120, it promoted paradoxical activation in *RAS* mutant cells.“

Thus, our experiment does not do what we intended, which was to inhibit BRAF signaling. To avoid confusion, we have chosen to remove this figure from the manuscript.

8) In Figure 6B, is it a robust result that the AKT inhibitor MK2206 only decrease the surface localization of CDCP1 but not CDCP1 total protein level?

This discrepancy could be due to the difference between the western blot data and the flow cytometry in terms of the way the data was collected and normalized. In the case of the western blot, cells were lysed, and total protein amount was normalized by BCA quantitation prior to loading on the gel. In the case of the flow cytometry, expression level was quantified on a single cell basis. Thus, one possible explanation for this result is that AKTi treatment, which reduces translation, caused a global reduction in protein level in cells, but that CDCP1 expression relative to a housekeeping control was unchanged. To avoid confusion, we have removed this experiment from the manuscript.

In place of this, we have performed a broader experiment where we measured the expression profiles of 8 cancer cells lines treated with AKTi. This new experiment more thoroughly explores the impact of AKTi on CDCP1 expression as well as the expression of 6 other candidate RAS surface markers we identified. Interestingly, AKTi appears to cause CDCP1 to increase slightly (also seen in new Figure 3). This perhaps could be due to a feedback mechanism whereby cells compensate for AKT inhibition by increasing signaling flux through ERK. This data is discussed in more detail in response to question 11.

9) NCI-H1299 cells have wild-type KRAS and NRAS-Q61K mutation, not KRAS-Q61 as stated in the text and Figure 6C. More RAS-mutant cell lines should be tested to further support the claim that CDCP1 expression is coupled with activation of MAPK signaling.

We thank the reviewers for noticing our mistake. The NCI-H1299 label has been corrected to note that the oncogenic mutation is in NRAS and not KRAS.

In addition, we have included more support in Figure 3 for the claim that CDCP1 is coupled to MAPK signaling activation. Briefly, we performed experiments in an additional 6 cell lines, where in 5/6 cases we observed dramatic reduction of CDCP1 expression levels after treatment with subtoxic concentrations of MEK inhibitor (PD0325901 100 nM). This is discussed in more detail in response to comment 11.

10) The authors have used publicly available databases to assess whether CDCP1 is overexpressed in KRAS mutant cancers. It would be helpful to determine this by a different method than probing gene expression (mRNA) as this is not entirely reliable.

We agree with reviewers that probing mRNA expression is not entirely reliable as we see above. We parsed through all available Pancreatic cancer datasets on the MSKCC Cancer Genomics Data Server, but there is a dearth of datasets featuring quantitative protein abundance measurements associated with genome sequenced cancers. This lack of proteomics data motivated us to profile the KRAS-induced cell surfaceome.

11) The experiment described in Figure 3C is not very informative given that no RAS WT cell lines are shown. Moreover, without some sort of KRAS perturbation (knockdown or MEK inhibition), there is no evidence that the expression of any of these proteins has anything to do with RAS mutation status.

We agree with the reviewers that more compelling evidence would help to support the notion that the expression observed in Figure 3C is KRAS signaling dependent. We have now included a RAS WT cell line, HEK293T cells, for comparison in Figure 3B. We found little detectable expression for all 7 targets probed.

While comprehensive expression profiling of hundreds of cancer cell lines would more thoroughly address the second point made by the reviewer, it is unfortunately outside the scope of what can be accomplished here. As an alternative, we heeded the reviewers’ suggestion to perturb KRAS signaling in the cells we measured through treatment with pharmacological inhibitors (MEKi, AKTi, EGFRi) as shown in the new Figure 3D mentioned above. Interestingly, we observed that several of these proteins (CD55, NT5E, CDCP1, and ICAM1) had expression that was responsive to KRAS pathway perturbation in mutant RAS cancer cell lines. Importantly, inhibition upstream of KRAS by EGFRi had little effect in most cases. We believe this data strengthens our claim that the observed expression of our panel of proteins was influenced in expression level by oncogenic KRAS signaling.

12) The type of data shown in Figure 5A is not very informative from a therapeutic point of view. Even though KRAS mutant and WT are statistically different at the population level, in truth only 7% of PDAC over-express CDCP1. 91% of PDAC (in the TGCA dataset) have mutant KRAS, so, regardless of the claims made in the paper, the general relationship between mutant KRAS is unclear. The Discussion should probably be more frank about this. Moreover, Figure 1F shows a less than impressive correlation between transcriptomics and proteomics, yet they are trying to make an important conclusion about CDCP1 protein expression from transcriptomics data on primary tumors. It would be useful if they could comment specifically on the correlation between RNA and protein for CDCP1 in their original analysis.

We agree that while the TCGA data we present does reach the level of significance, the effect size is rather modest. As we have tried to articulate in the paper, we believe that RNA expression level may not be strongly predictive of protein abundance, and thus the direct link between KRAS mutational status and CDCP1 in PDAC needs further experimental support. While we are excited to further explore this link, we believe it goes beyond the scope of this manuscript to generate this data broadly. Hence, we have removed Figure 5A from the manuscript and used different language to articulate our exploration of CDCP1 in PDAC in the text:

“Oncogenic KRAS is nearly ubiquitously expressed in pancreatic cancers, and these cancers are particularly dependent on RAS signaling through the MAPK pathway (Eser et al., 2014). Thus, we profiled CDCP1 expression levels on a panel of human […]”

Deleted text

“We thus looked into transcriptomics data in pancreatic cancer derived from The Cancer Genome Atlas (TCGA), which revealed that CDCP1 transcript abundance was significantly upregulated and correlated with KRAS mutational status (Figure 5A).”

[Editors' note: further revisions were requested prior to acceptance, as described below.]

The manuscript has been improved but there are some remaining issues that need to be addressed before acceptance, as outlined below:The Abstract, which emphasizes biological findings, remains unchanged. For example, the statement, "we discovered a signature of proteins critical for metastasis that are upregulated on cells transformed with KRASG12V, and driven by MAPK pathway signaling." is misleading in light of the findings presented. Similarly, the phrasing of the statement describing CDCP1 findings still leads readers to conclude that this protein has been newly identified by these studies. The authors need to edit the Abstract to appropriately represent the key novel claims, and the emphasis on the new technology.

We appreciate this comment and have modified the Abstract in response. To address the sentence in question, we have removed “critical for metastasis” since this is not directly shown in our work here. We have also edited the sentence mentioning CDCP1 to emphasize that it can be targeted for ADC or T-cell recruitment strategies. We also emphasize the technology platform as a key development in the final sentence. The edited Abstract is below:

“While there have been tremendous efforts to target oncogenic RAS signaling from inside the cell, little effort has focused on the cell-surface. Here, we used quantitative surface proteomics to reveal a signature of proteins that are upregulated on cells transformed with KRAS^G12V^, and driven by MAPK pathway signaling. […] We show that antibodies targeting CDCP1, a protein common to our proteomics and CRISPRi datasets, can be leveraged to deliver cytotoxic and immunotherapeutic payloads to RAS-transformed cancer cells and report for RAS signaling status in vivo. Taken together, this work presents a technological platform for attacking RAS from outside the cell.”

The authors have not adequately addressed a concern that was raised regarding the transition from the CRISPRi discovery of a requirement for CDCP1 and the experiments in which it was targeted using an ADC strategy (subsections “Functional characterization of the KRAS surfaceome in MCF10As using a CRISPRi screen” and “Antibodies can selectively deliver toxic and immunotherapy payloads to mutant KRAS pancreatic cancer cells”. In the reply to reviewer comments, the authors refer to a statement they make where they write: "One of the most common mechanisms of resistance to targeted therapies is loss of expression of the protein target." But this is not true. In fact, most targeted therapies are directed to proteins that are required by cancer cells and such targets are therefore not typically reduced in expression as a resistance mechanism. The authors should remove or restate this.

It is not essential that we make these points to rationalize the CRISPRi screen. Thus, as suggested by the reviewer, we have deleted the first three sentences of that paragraph and now begin the paragraph as follows:

“In addition to identifying proteins upregulated by expression of oncogenic KRAS, we explored how oncogenic KRAS signaling can influence the dependence of cells on surface proteins.”

More notably, the authors should explain in the transition between these two sections that the dependency of RAS-transformed cancer cells on CDCP1, together with its high surface expression, makes it an attractive ADC target-which is unlikely to be selected against as a resistance mechanism due to its requirement in these cancer cells. Therefore, they explored the ADC approach. They should also comment on the fact that their antibody to CDCP1 is apparently not a function-blocking antibody-since it does not seem to affect cell viability.

We have modified the manuscript to strengthen the transition between these sections, and point out that the antibodies are not functional inhibitors. We have modified the manuscript as follows to address this:

“While our results provide a large number of interesting proteins for follow-up, we decided to focus on targeting CDCP1 with our validated recombinant antibodies. […] These qualities suggest that CDCP1 could be a particularly attractive target for an antibody drug conjugate (ADC).”